# Carbon accumulation in recently deposited peat is reduced by increased nutrient supply

Betty Ehnvall [1,2] ✉, Joshua L. Ratcliffe[1,3], Carolina Olid [4], Jacob Smeds[1], Kevin Bishop [2], Jonatan Klaminder[1], Chuxian Li[2,5], Carl-Magnus Mörth [6], Mats B. Nilsson [1] & Mats G. Öquist [1]

High-latitude mires store a considerable part of the global soil carbon. Current understanding suggests that wetter conditions promote carbon accumulation. This paradigm is based primarily on temperate ombrogenic bogs and overlooks the influence of minerogenic water from the catchment area, despite most northern mires being minerogenic fens. Here we show that minerogenic water is the main negative influence on past century carbon accumulation in boreal fens. This effect is most pronounced in mires formed during the last millennia. Rather than enhancing productivity, minerogenic water stimulates organic matter decay, apart from in elevated hummocks where both decay and productivity were stimulated. These findings reshape our understanding of carbon cycling at high-latitudes, highlighting how shifts in precipitation-evapotranspiration may impact carbon sequestration in fens, which are widespread in the circum-arctic. Contrary to expectations for temperate regions, we argue that increased catchment water input in sub-arctic peatlands is unlikely to enhance mire carbon accumulation.

The importance of northern mires, i.e. peat-forming wetland ecosystems, in the global carbon (C) cycle is primarily attributed to their ability to sequester and store C at geological, i.e. multi-millennial, timescales[1,2]. From this long-term perspective, C accumulation in mires removes carbon dioxide ($CO_2$) from the atmosphere, resulting in a net-negative radiative forcing outweighing the effects of methane ($CH_4$) emissions[1]. The water table and its influence on oxygen availability is currently believed to be the main enabler of peat accumulation, where long-term (annual to millennial) C storage is a result of very slow mass loss in the deep and permanently water-saturated anoxic peat layer ('catotelm') characterized by slow anaerobic decomposition[3]. Importantly, C accumulation over an annual to decadal timescale first occurs in the unsaturated surface zone ('acrotelm'), where decomposition rates are higher due to readily available oxygen as well as an often high proportion of poorly decomposed labile C in the peat. The net annual atmospheric C exchange from mires is often very small and can occasionally shift from being sinks of atmospheric C to becoming sources, where reduced C retention is almost always related to periods with deeper groundwater levels and more oxygenated conditions in the surface peat[4,5].

In addition to the water table, nutrient availability is a critical factor in determining the type of peat-forming plant community that will be established[6]. Plant communities ultimately affect the accumulation of C in mires due to different sensitivity to environmental drivers and different decomposition rates of their litter. Some species also directly engineer the ecosystem, altering peat forming factors such as temperature and moisture. For example, *Sphagnum* mosses are capable of promoting a shallow and stable water table through various

[1]Department of Forest Ecology and Management, Swedish University of Agricultural Sciences, Skogsmarksgränd 17, 90183 Umeå, Sweden. [2]Department of Aquatic Sciences and Assessment, Swedish University of Agricultural Sciences, 75007 Uppsala, Sweden. [3]Unit for Field-Based Forest Research, Swedish University of Agricultural Sciences, 922 91, Vindeln, Sweden. [4]UB-Geomodels Research Institute, Departament de Dinàmica de la Terra i l'Oceà, Facultat de Ciències de la Terra, Universitat de Barcelona, 08028 Barcelona, Spain. [5]Institute of Geography and Oeschger Center for Climate Change Research, University of Bern, 3012 Bern, Switzerland. [6]Department of Geological Sciences, Stockholm University, Svante Arrheniusväg 8, 10691 Stockholm, Sweden. ✉e-mail: betty.ehnvall@slu.se

hydro-physical feedbacks[7], and this group of mosses are central for peat accumulation in high-latitude mires[8,9]. The productivity of *Sphagnum* is often limited by nitrogen (N) availability[10–12] in regions with low atmospheric N deposition[13]. In N-limited systems, vegetation relies on the N fixation capacity of the mosses and the rates of N mineralization within the peat to support N availability. In minerogenic, i.e. groundwater-fed mires (fens), which are the dominant mire type at high latitudes[14], N fixation can be stimulated by the availability of weathering-derived nutrients from the catchment[15], primarily through the supply of phosphorus, but also by iron and molybdenum[16]. In this way, water inputs from the catchment area can be expected to affect both the C and N accumulation rates in mires.

At the local spatial scale of a single mire, the presence of microtopographic features, such as elevated hummocks and the more extensive, flatter lawns introduces additional complexity as these microtopographic features represent a spectrum of wetness that influences oxygen and nutrient availability within the mire[17]. For example, hummocks tend to be drier, with more porous peat and lower bulk densities than peat found in wetter microforms. Seasonality in climate and magnitude of precipitation and evapotranspiration can also affect nutrient transport within the mire, causing differences in nutrient availability across microtopographic features, where hummocks typically get higher nutrient inputs during dry conditions, while lawns get a higher nutrient load during periods of high groundwater availability[17]. This variation in peat characteristics can result in significant differences in peat C and N accumulation between hummocks and lawns within the same mire[18–20].

Peat formation over millennial time-scales has served as a foundation when developing existing theories on the spatio-temporal factors controlling C accumulation rates at both individual mires[21–23], and at larger regional scales[3,24–26]. When considering specifically the processes leading to long-term C accumulation, it is crucial to focus on the oxic acrotelm where most of the primary production and decay occur. It is well known that near-surface C accumulation rates are higher in comparison with in deeper peat layers as peat becomes progressively older and thus, more degraded with depth[27,28]. However, these accumulation rates can also represent varying productivity and decay dynamics, with different implications for the C input into the relatively stable, anoxic catotelm[29]. For instance, increased decay and productivity may increase long-term C inputs in some sites while reducing them in others[30]. Therefore, interpreting C accumulation rates requires careful consideration taking into account both organic matter decay rates and the acrotelm residence time. By examining the acrotelm, we can better relate present environmental conditions, including morphological, geochemical and climatic conditions, to contemporary surface peat rather than peat formed thousands of years ago[31]. Hence, understanding the drivers of recent peat accumulation rates is key as they have major implications for the contemporary C balance of high-latitude mires and the C currently entering or soon to enter long-term storage in the catotelm.

Here, we demonstrate how recent (i.e. past century) peat, C and N accumulation rates, along with peat productivity and organic matter decay rates, vary over a 5000 year mire initiation age gradient using a mire chronosequence in a constrained area with similar climatic conditions. The unique landscape settings formed by post-glacial isostatic land-uplift allow us to disentangle the role of mire succession (i.e. landscape ageing), from geomorphological catchment controls on mire nutrient regimes and accumulation patterns. More specifically, we unravel (i) how recent peat, C and N accumulation rates have changed over ~5000 years of mire development, (ii) how accumulation rates vary in response to catchment hydrogeochemistry and nutrient availability, and (iii) how peat, C and N accumulation rates and their responses to spatio-temporal drivers differ between microtopographic features at the mire surface.

## Results

We studied recent mass accumulation rates of peat (MAR: 170–280 g m$^{-2}$ yr$^{-1}$), carbon (CAR: 85–140 g C m$^{-2}$ yr$^{-1}$) and nitrogen (NAR: 1.1-2.0 g N m$^{-2}$ yr$^{-1}$) over the past century using $^{210}$Pb chronologies. Over the entire mire population, CAR was up to 75 g C m$^{-2}$ yr$^{-1}$ higher in lawns than in hummocks. To identify the drivers behind changes in accumulation rates, we evaluated various mire and catchment properties as explanatory variables. Average long-term peat accumulation rates based on the total peat depth would be difficult to interpret due to oligotrophication and/or ombrotrophication, with subsequent changes in peat accumulation rates, that occur across the age gradient. Total peat depth also does not necessarily reflect the rate of peat accumulation, as peat porosity varies between sites, usually with more porous peat in the younger mires. Therefore, we expect that recent accumulation rates will better reflect differences in peat accumulation across the trophic gradient than long-term average accumulation rates.

Neither mire age, peat depth, nor water table depth alone could directly explain changes in peat accumulation rates (Fig. 1), suggesting that ecosystem ageing per se is not the primary driver. However, in hummocks, accumulation rates were indirectly influenced by mire age and the build-up of the hummock microform, as evidenced by the strong contribution of age-related variables (age, catchment elevation, peat depth, mire area, catchment area) to the orthogonal projections to latent structures model (OPLS). In contrast, mire age did not affect accumulation rates in the more spatially dominant lawns (Fig. 2). Furthermore, mire age and water table depth were not correlated, indicating that the observed temporal patterns associated with mire age cannot be attributed solely to differences in water table levels or acrotelm depth (Fig. 3). This suggests that additional factors must also play a role in shaping the dynamics of peat accumulation.

We propose that nutrient availability from weathering processes enhance peat decay in the youngest minerogenic mires studied, resulting in a decreasing trend in lawn decay rates (based on Clymo's[29] model) with increasing mire age (Fig. 4a, $R^2 = 0.9$, $p = 0.01$). Elements in peat can promote peat decay both directly, by participating in electron transfer, and indirectly as nutrients influencing plant community composition. Higher nutrient support from the catchment in the studied mires was reflected in higher productivity in hummocks and higher vegetation greenness (Fig. 3), in agreement with previous studies in the area[15].

Mineral soils in catchments closer to the present coastline of the isostatically rising land have been exposed to weathering for a shorter period of time compared to soils further inland[32]. This has profound implications for the composition and flux of nutrients exported to mires across the age gradient[15]. As the landscape ages, lateral inflow of nutrients to mires declines, and vertical nutrient inputs decrease with increasing peat depth[33]. Consequently, peat-forming vegetation in older mires must rely more on recycled nutrients, which are depleted over time as nutrients become buried and immobilized in partially decomposed organic matter[34]. Atmospheric deposition may act as a background source of nutrients, in addition to inputs from soil or water. Given the proximity of the youngest and oldest mire (<20 km), deposition rates are likely to be relatively uniform across the chronosequence area, playing potentially a greater role in older mires compared to the younger mires, which are also fed by weathering products[15]. Interestingly, N deposition can potentially promote peat decay in shrub-covered mires, with consequent peat C losses, due to decreased ericoid mycorrhizal activity in peatlands[35].

Interpreting recent, i.e. acrotelm, MAR and CAR requires special consideration with regards to whether the dynamics we record here will persist to the long-term C storage pool in the catotelm peat[28]. For instance, Ohlson and Økland[20] showed that the top 20 cm of peat could represent anywhere from 7–173 years of peat accumulation. It follows that sites with a short acrotelm residence time of less than or a

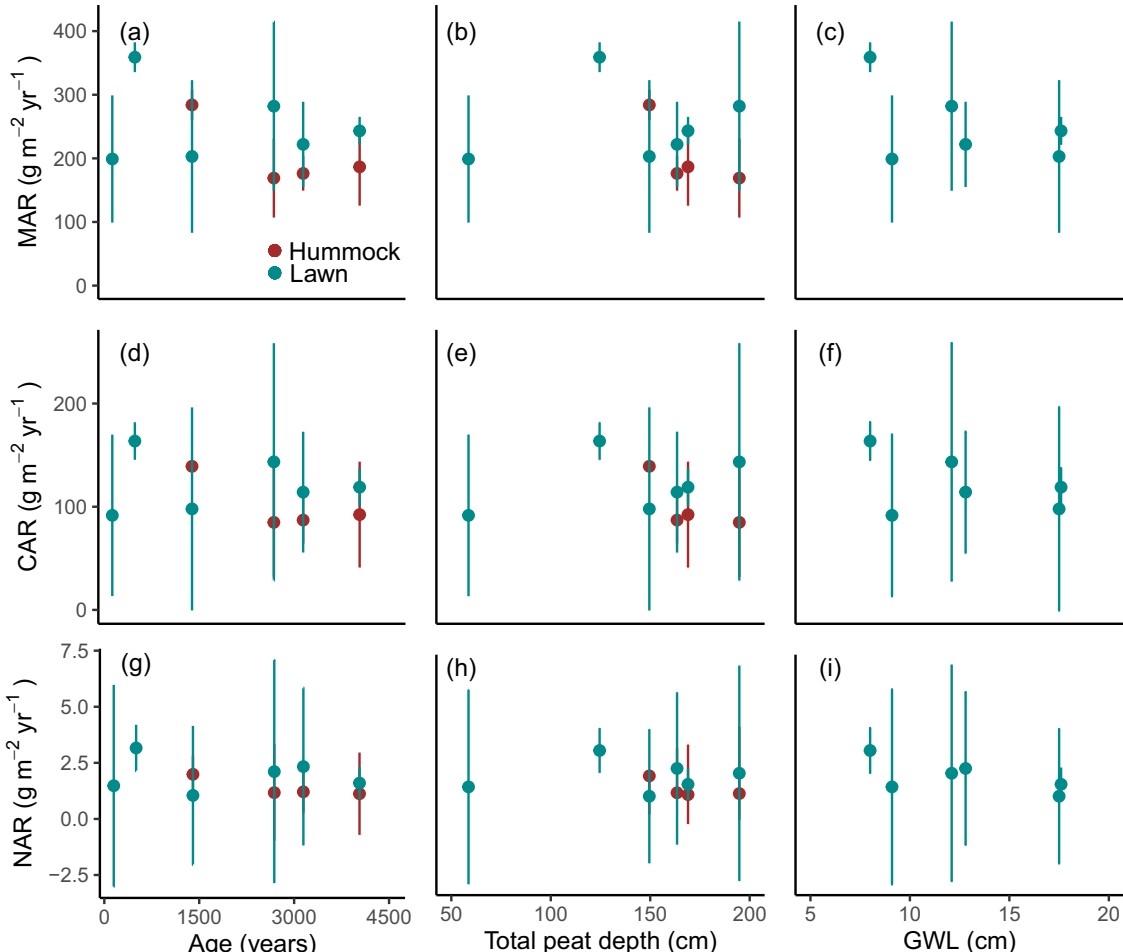

**Fig. 1 | Peat mass (MAR), carbon (CAR) and nitrogen (NAR) accumulation rates in lawns (blue) and hummocks (red).** Accumulation rates are compared to mire ages (**a**, **d**, **g**), peat depths (**b**, **e**, **h**), and median water table depths (GWL) over the vegetation period (**c**, **f**, **i**). Error bars correspond to standard errors from the ²¹⁰Pb-based modelling, as well as in the case of CAR and NAR from analytical uncertainty related to carbon and nitrogen mass fractions. In (**c**, **f**, **i**) no hummocks are displayed due to the lack of water table depth monitoring data from these microforms.

few decades will be relatively insensitive to the rate of decomposition, in contrast to sites with a long acrotelm residence time of up to a century, where aerobic decomposition has more time to act and therefore more influence, leading to greater C losses[29].

At our sites, peat from ~1920 was found between 29 and 47 cm depth, well below the average water table in all cores, representing the catotelm. After 100 years of decay, an average of 41% of the productivity was found to remain (range: 14%–89 %, data in repository), indicating active peat accumulation in the anoxic catotelm at a mean rate of ~50 g m⁻² yr⁻¹. Doubling the decay rate across all sites, which is comparable to the difference in decay rates between the lawns in the oldest and youngest mires, resulted in an average 52% reduction in C remaining after 100 years, with individual cores showing reductions between 11% and 86%. These variations in recent MAR, CAR and decay rates are, therefore, expected to drive changes in the long-term C storage of the catotelm.

MAR and CAR have earlier been proposed to be negatively affected by variables associated with minerogenic water inputs, such as surface peat concentrations of Si, Fe and Mg[36]. Concentrations of these elements co-varied with the catchment-to-mire ratio (C-to-M; Fig. 3), suggesting that large catchments contribute with more weathering-derived elements to recipient mires. Our results indicate that hummocks are more sensitive to nutrient availability, mire size, and ageing compared to lawns whereas lawns are more influenced by catchment size (Fig. 2).

Based on our OPLS models, the dynamics of lawns follow our hypothesis: higher decay rates covary with lower MAR, CAR, NAR, while higher primary productivity covaries with higher accumulation rates. Looking at productivity and decay rates separately (Fig. 4), it is clear that lawn decay rates decline as mires ages, likely due to reduced minerogenic nutrient inputs (Fig. 4a, $R^2 = 0.9$, $p = 0.01$). In contrast, lawn primary productivity remains constant over time (around 270 g m² yr⁻¹), with no clear dependence on mire age or water table depth (Fig. 4). This suggests that nutrient availability alone is not the main driver of lawn peat productivity. Because of the high productivity rate compared to total mass loss, nutrient conditions have not significantly altered MAR in lawns over the past 100–150 years (Fig. 2). In addition, the residence time of peat in the acrotelm is likely to be shorter in lawns than in hummocks due to a generally shallower water table, which may further limit the impact of peat acrotelm decay in lawns compared to hummocks.

## Discussion

By applying the mire chronosequence approach, we can bridge the gap between recent and long-term peat accumulation rates and examine how recent peat accumulation rates change as the mire ages, which is an important complement to studies that focus solely on either long or short-term accumulation rates. By focusing on the past 100–150 years our data are much easier to compare with morphological, geochemical, and climatic data than typical paleo-chemical studies, where it is

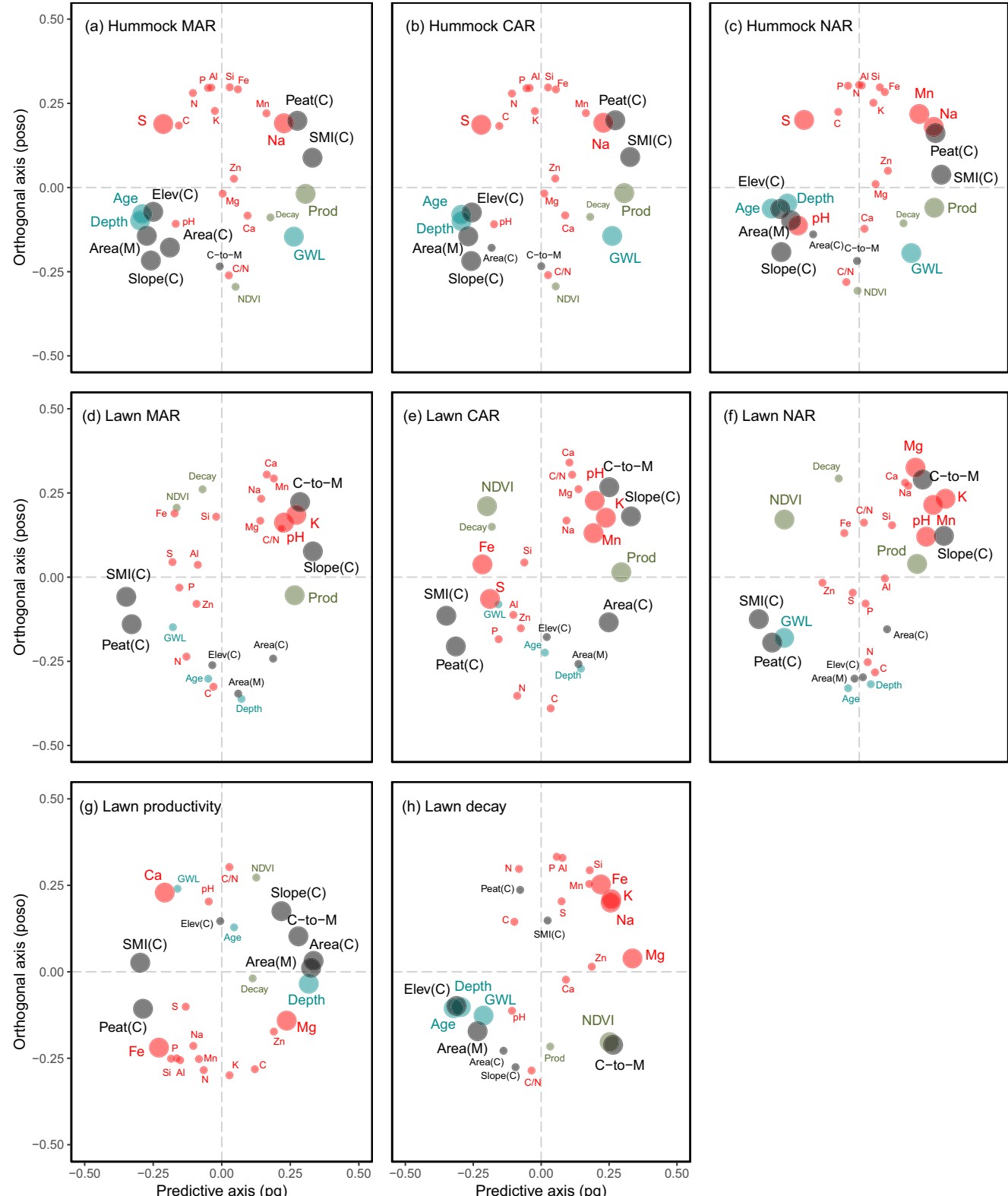

**Fig. 2 | Orthogonal projections to latent structures models (OPLS) used to predict peat mass (MAR), carbon (CAR) and nitrogen (NAR) accumulation rates in hummocks and lawns.** Panels correspond to (**a**) mass accumulation rate in hummocks, (**b**) carbon accumulation rates in hummocks, (**c**) nitrogen accumulation rates in hummocks, (**d**) mass accumulation rates in lawns, (**e**) carbon accumulation rates in lawns, (**f**) nitrogen accumulation rates in lawns, (**g**) peat productivity in lawns and (**h**) peat decay rates in lawns. Each model contain one predictive and one orthogonal component. Predictors are grouped based on type of predictor into peat elements (red), catchment characteristics (grey), peat growth (green) and physical peat properties (blue). Predictors with highest absolute values at the predictive axis, and closest to the orthogonal axis, contribute strongest to the OPLS model. Predictors that according to their predictive VIP scores contributes significantly to the predictions are marked with large circles, while predictors that were not significant are marked with small circles. Abbreviations are explained in Supplementary Information's Table S3.

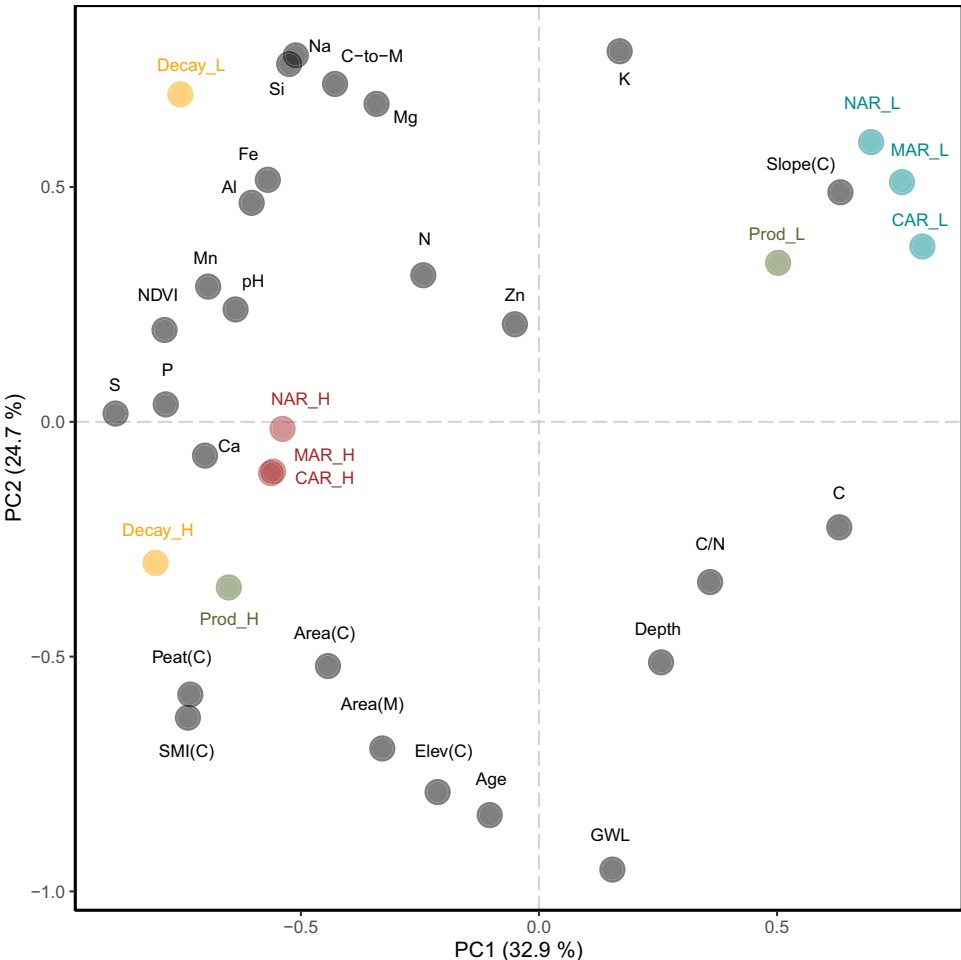

**Fig. 3 | Principal component analysis (PCA) covering peat mass (MAR), carbon (CAR) and nitrogen (NAR) accumulation rates.** Accumulation rates in lawns (L) are marked in blue and accumulation rates in hummocks (H) are marked in red, while environmental variables (are marked in grey).

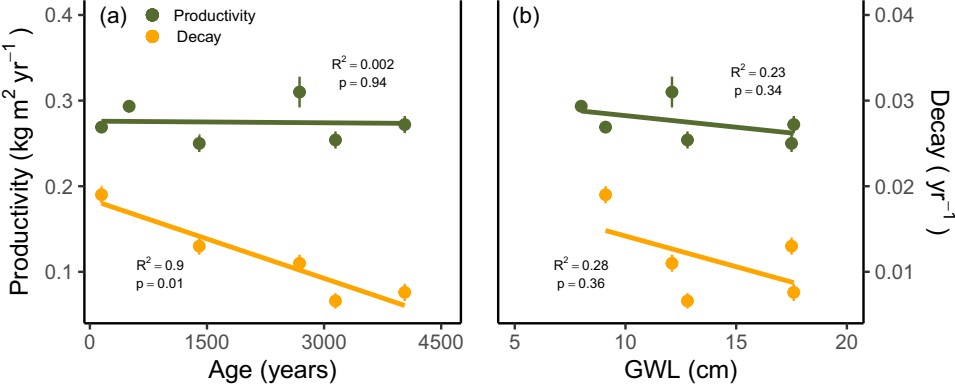

**Fig. 4 | Peat productivity (green) and decay rates (orange) in lawns.** Rates are derived from Clymo's model on peat accumulation[29], and are reported relative to mire age (**a**) and water table level (**b**). Peat productivity refers to the gross peat formation, i.e. the addition of new peat by the mire surface, while the decay refers to the decay constant. Error bars correspond to uncertainty of Clymo's model outcome.

not uncommon for present day explanatory data to be paired with CAR over thousands of years or more[31].

Our results from lawns are consistent with the theory that C accumulation in northern mires is limited by a stronger influence of groundwater relative to rainwater[36–40]. Contrary to what might be expected for ombrogenic mires (bogs), which receive water and nutrients only from precipitation, reductions in minerogenic water inputs in minerogenic mires (fens), can promote vigorous growth of *Sphagnum* mosses and trigger the so-called fen-to-bog transition[41]. This 'chemical' limit on C accumulation[36], caused by an excess of groundwater and associated nutrients, is quite different from the classical theories of peat accumulation put forth by Clymo[29,42] and Ingram[43], which focus on the effect of water balance on raised bog growth in temperate climates. Following these strictly hydro-climatical

models, greater water inputs always lead to deeper peat formation. It has long been suspected that groundwater inputs, and possibly through-flow[44] and the removal of phenolic compounds[44], limit peat and C accumulation[45,46]. In the boreal region, this has even been proposed as a major mechanism that is more important than climatic wetness for limiting the eventual 'ombrotrophication' of boreal mires[37,38,47]. Our main finding is that catchment water input limits CAR in lawns by stimulating peat decay. Furthermore, both productivity and decay terms are stimulated by nutrients in hummocks[48], but because hummocks are less abundant compared to lawns, the net effect on C accumulation at the ecosystem level is neutral. Contrary to what might be expected, hummock decay rates were more strongly driven by the catchment support of nutrients compared to lawns. Our findings highlight the need of future research to evaluate the magnitude and dynamics of nutrient inputs to better understand C accumulation in mires.

Our results emphasize the role of groundwater in limiting peat accumulation in boreal mires. This is a highly important finding because the negative effect of groundwater on peat decay contradicts the widely held assumption that wetter conditions lead to greater C accumulation and vice versa[49]. What is more, this finding would be completely overlooked by most studies on peatland CAR, both paleo and contemporary, which focus almost exclusively on first-order climatic variables. Our research also highlights the link between water export from the catchment and the balance between C accumulation and loss in mires. Land-use that impacts water and nutrient export from the upland areas to the mire, for example changes in evaporation or groundwater disturbance associated with forest management, could affect peat decay and C exchange elsewhere in the catchment. Thus, changes in land-use and the precipitation-evaporation of upland areas are most likely to impact the decay rate of surface peat and the C accumulation rate across the dominant lawn microtopographic feature, which covers the majority of the mire surface in high-latitude minerogenic mires. Our work adds evidence pointing towards catchment water inputs as a primary control of peatland C balance in high-latitude mires.

## Methods

### Regional settings

The study was conducted in the Bothnian Bay Lowlands (BBL) in the county of Västerbotten in northern Sweden. The region was covered by the 3 km thick Scandinavian Ice Sheet during the last glacial period[50] and, as a result, represents one of the areas globally with highest land-uplift, today at a rate of around 9 mm yr$^{-1}$ [51]. The continuous exposure of new land areas from the Baltic Sea creates suitable conditions for mire initiation[52], which has formed a natural mire age-gradient between the present coastline and the highest coastline (around 250 m above sea level). This age-gradient, or chronosequence, can be used to study long-term mire development at the landscape-level[53,54].

The retreat of the Scandinavian Ice Sheet has formed a characteristic, undulating landscape mosaic of elongated, wave-exposed and till-covered ridges (drumlins) interlaid by valleys of postglacial silt, clay and sand[55]. Paragneiss (quartz, feldspar and mica), dominates the bedrock in the area, but felsic (granodiorite, granite) and mafic rock intrusions (basaltic andesite, gabbrodiorite) are present, according to the Geological Survey of Sweden (Bedrock maps, 1: 50,000). Annual average temperature and precipitation in the area is 3.5 °C (July 15.7 °C, January −6.8 °C) and 654 mm (July 79 mm and January 48 mm) based on 30 year mean values from the meteorological stations closest to the study area (Umeå airport and Vindeln-Sunnansjönäs; The Swedish Meteorological and Hydrological Institute, 2022). The coastal-most parts of the study area receive around 10% more solar radiation (around 100 kWh) over the growing season (above 3 °C) compared to the more inland parts.

### Study sites

The study covers eight minerogenic mires in the Sävar Rising Coastline Mire Chronosequence (SMC; Fig. 5). The SMC mires are located within 15 km from the present coastline and span an altitude range of 1.5−54 m above sea level (m a.s.l.), corresponding to a land surface age range of around 0−4500 years (Table S1)[56]. Photos from the sampled chronosequence mires are found in our interactive mire chronosequence map (https://slughg.github.io/MiresChrono).

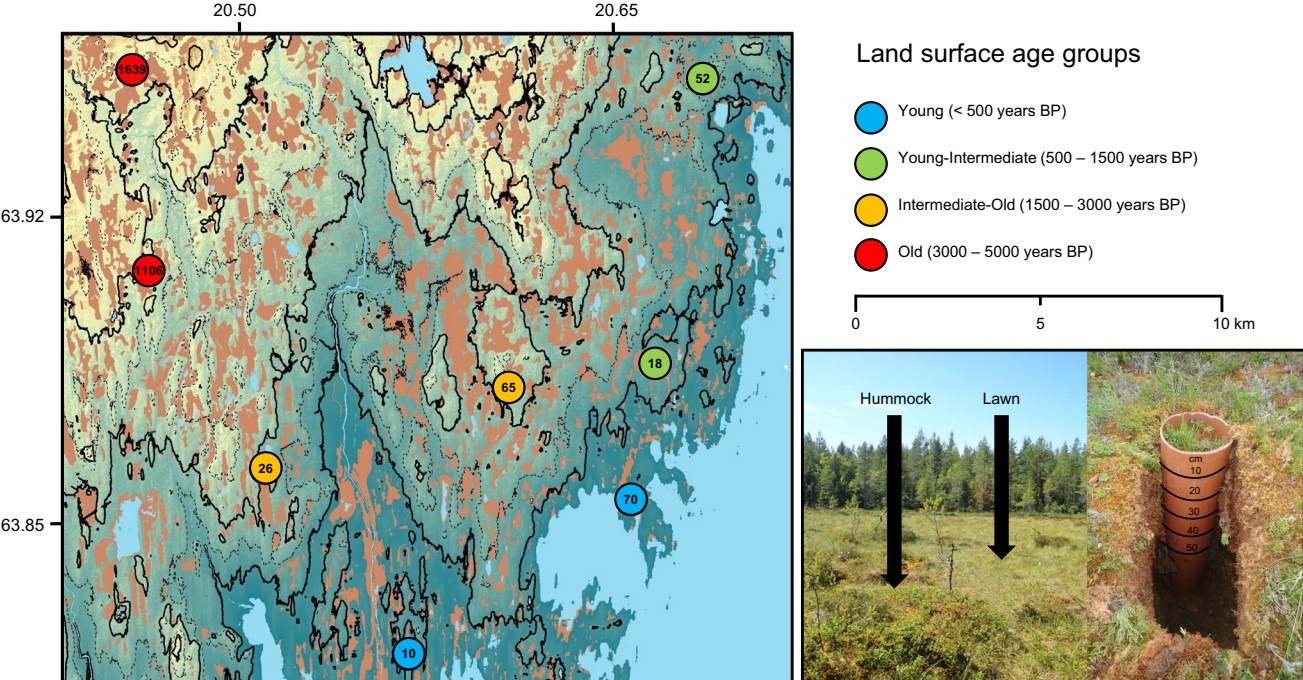

**Fig. 5 | Map of study sites.** Eight mires in the 0–5000 years age range in the Sävar Rising Coastline Mire Chronosequence (SMC) were included in the study. The mires are found in coastal areas of the Swedish part of the Bothian Bay Lowland (BBL) in northern Baltic Sea. 50 cm peat cores were collected from two topographic microforms; lawns and hummocks.

## Peat sampling and sample preparation

Hummock and lawn cores were collected from the mires in July–August 2019 (Mire ID S10, S18, S26, S52, S65, S70) and in July 2020 (Mire ID 1106 and 1639; Table S1). At one of the youngest sites (S10), hummocks had not developed and, hence, a single lawn peat core represented the micro-topographically homogenous mire surface. Peat cores, including the uppermost living plants, were sampled using a 50 cm deep cylindrical peat corer with an inner diameter of 15.1 cm (modified from Clymo[57]) and with the mire surface as the reference level. Due to shallow peat depth, cores from one of the youngest mires (S70) were only 26 cm (hummock) and 30 cm (lawn) deep. Peat cores from the two oldest mires (1106 and 1639) were sampled from the centre of the mire according to Ehnvall et al.[15], while cores from the remaining mires (S10, S18, S26, S52, S65 and S70) were collected from locations previously used in studies such as Wang et al.[58].

Peat cores were stored in air- and watertight PVC tubes sealed with plastic caps and transported to a freezer (−18 °C) within 6 h after extraction. The cores were cut into 2 cm thick layers within a month after sampling, using a band saw with a stainless-steel blade (Metabo Bas 318). To generate undisturbed layers, we cut the peat in temperatures below +4 °C. Discs from 11 cores were oven-dried at 60 °C for 96 h, while discs from four cores were freeze-dried[59]. To avoid moisture adsorption, the oven-dried samples were cooled down in a desiccator. Dry bulk density (BD) was then measured gravimetrically based on one-half of the 2 cm disks. The dried samples were homogenized and ground at 10 s intervals using a tube mill with rotating speed of 2500 rpm (IKA Tube Mill Control, version 1.4). Peat organic matter content (OM%) was calculated after drying samples at 105 °C overnight, followed by loss on ignition (LOI) at 550 °C for 6 h.

## Estimating mean peat, carbon and nitrogen accumulation rates

Accurate recent (~100 years) peat chronologies of the collected profiles were obtained from the distribution of the natural radionuclide $^{210}$Pb ($T_{1/2} = 22.3$ yr)[60]. Total concentrations of $^{210}$Pb were determined through the analyses of its granddaughter $^{210}$Po, which was assumed in secular equilibrium with $^{210}$Pb. From each peat layer, a 0.1 – 0.2 g subsample was acid digested (HNO$_3$, 9 mL) using an analytical microwave, followed by spiking with a known amount of $^{209}$Po yield tracer[61]. After this, the material was plated onto silver discs and measured by alpha spectrometry using Ortec ULTRA-AS Ion-Implanted-Silicon Charged-Particle Detectors (Model U-020-450-AS). Concentrations of unsupported or excess $^{210}$Pb ($^{210}$Pb$_{xs}$) used to obtain the age-depth models were determined as the difference between total $^{210}$Pb and supported $^{210}$Pb. Supported $^{210}$Pb was taken as the concentration of $^{210}$Pb in the deepest part of the core, where the $^{210}$Pb profile becomes invariant with depth. On estimating accurate ages due to the potential downward mobility of $^{210}$Pb in the uppermost peat layers[62,63], profile chronologies were determined based on $^{210}$Pb using the Constant Flux:Constant Sedimentation (CF:CS) model[64] or the Constant Rate of Supply (CRS) model, depending on which model provided an accumulation rate that was more consistent with the SCP record[65]. The CF:CS model could be applied in four out of the total fifteen sampled peat cores. Of these, one core was a lawn and three hummocks. For five lawn cores and one hummock core, the CRS model was applied, meaning that at the end 10 out of 15 cores could be dated and included in the statistical analyses. In S70, S10 and S52 peat cores did not reach the unsupported fraction of $^{210}$Pb, either due to shallow total peat depth below 50 cm (S70) or because the mires accumulated peat faster than expected, which meant that a 50 cm peat core was not deep enough to reach the unsupported background levels of $^{210}$Pb. At these sites, the deepest peat layer corresponded to years 1952 ± 20 yrs (S70 lawn), 1960 ± 20 yrs (S10) and 1964 ± 20 yrs (S52 lawn).

The $^{210}$Pb profiles showed significant variations from the expected exponential decay, requiring an independent chronological marker to validate the $^{210}$Pb-derived chronology[62]. In addition to the $^{210}$Pb dates, we used the vertical distribution of spheroidal carbonaceous particles (SCP) in the peat record to infer an independent, supporting, chronological marker in line with techniques described in detail elsewhere[66]. In short, SCP is dependent on temporal increases in the atmospheric fallout that is reflected as increased concentration of SCPs in sedimentary records. Chronologies using SCP have been outlined for the study region and used to identify peat deposited around the 1980s[67]. SCP was analyzed from 100 mg dried, homogenized peat after acid digestion (HNO$_3$, 5 mL and H$_2$O$_2$, 1 mL) using Mars Microwaves (CEM Corporation, Charlotte, NC, USA). The SCPs were counted after drying the samples using microscopy (50 times resolution). Cores analyzed for SCP were S70L, S70H, S10, S52H, S52L, S18L, S18H, S26L, S26H, as well as S65H.

Annual C inputs and peat decomposition rates were estimated by fitting the cumulative C stocks versus derived-ages to the general equation provided by Clymo's model for peat soil[29]. This simple model treats the accumulating peat as a two-layer system: the acrotelm (the unsaturated, oxic, upper layer) and the catotelm (the underlying, permanently saturated, anoxic peat layer), with the boundary of these two layers at the maximum water table depth. This model assumes that the net change in the C storage in a peat deposit is given by the imbalance between annual C inputs (productivity $P$) and decomposition losses ($kC$):

$$\frac{dC}{dt} = P - kC \tag{1}$$

When solved, the temporal variation in the cumulative C is given by:

$$C(t) = \frac{P}{k}(1 - e^{-kt}) \tag{2}$$

Where $C(t)$ represents the cumulative C stock (g C m$^{-2}$) at a certain time $t$, P is the annual C input (g C m$^{-2}$ yr$^{-1}$) at the surface, and $k$ is the first-order decomposition constant (yr$^{-1}$).

The application of Clymo's model for estimating primary production and decay rates was primarily successful in the lawn cores. However, we found some difficulties when applying it to the hummocks, likely due to the non-homogeneous nature of the peat and the strong influence of water table fluctuations, which might have affected the retention of $^{210}$Pb and introduce uncertainties in the derived ages[62]. As a result, productivity and decay rates are reported solely for the lawns when considering trends over time.

Total mass fractions of C and N in each peat layer were measured on samples stored at −18 °C. All peat samples were homogenized, ground and re-dried at 70 °C for 18 h and cooled down in desiccator prior to measurements. After drying, 5 mg peat from each layer was weighted into a tin capsule and analyzed using an Elemental analyzer (Flash EA 2000, Thermo Fisher Scientific, Bremen, Germany). For each peat profile, the mass fractions were calculated based on the dry weight of peat. Mean mass accumulation rate (MAR) was multiplied by the mean C mass content to derive the mean C accumulation rate (CAR) over the past 10 years. Similarly, the mass accumulation rate was multiplied by the mean N mass-content over the profile to achieve the mean N accumulation rate (NAR) over the past 100 years. C and N concentrations over the peat profiles were also calculated by multiplying mass-ratios of C and N, respectively, by the organic matter content obtained by LOI and dry bulk density measurements.

## Validation of inferred carbon accumulation rates

The estimated recent CAR was relatively high compared to mires in boreal parts of Canada[68] (34–52 g C m$^{-2}$ yr$^{-1}$), but within the range of recent CAR estimated for Degerö Stormyr, which is located ca 60 km inland from the studied chronosequence[69]. This is interesting because studies of CAR based on mires representing early successional stages,

such as those considered in our study, are scarce due to the rare occurrence of young landscapes (<5000 years) with strong weathering gradients. Consequently, most studies of recent accumulation rates based on $^{210}Pb$ chronologies are based on ecosystems that have accumulated peat over much longer periods than the mires described here. The estimated NAR for the last hundred years was within the same range as the NAR in corresponding young landscapes in Finland[53] and in the northern-most parts of Sweden[70].

## Peat and water table depth

Total peat depth was measured at each sampling location by pushing a metal rod through the peat until it reached a non-penetrating surface of bedrock, rock or mineral soil. For sites 1106 and 1639 peat depth was measured within 5 m from the sampling sites[15] and for the younger chronosequence mires peat depth was measured within 50 m from the sampling points.

Each of the mires has been instrumented for continuous measurements of groundwater levels. The groundwater loggers in the SMC were installed in 2021 (sites S10, S18, S26, S52, S56, S70; although here we only use data from 2022-2023) and 2023 (sites 1106 and 1639) in screened PVC tubes with holes every 0.5 cm, allowing for the water table to equilibrate with the outside, and protecting the loggers from damage during operation[71]. In the six youngest mires, Odyssey® Xtreem Capacitance Water Level Loggers represented groundwater levels in lawns within 50 m from the peat coring sites, while in the two oldest chronosequence mires (sites 1106 and 1639) the Levellogger 5 Junior (Model 3001) represented lawns within 5 m from the peat coring sites. At site 1639, the Levellogger 5 Junior Barologger was used to measure the barometric pressure required to correct the groundwater levels monitored by the Levellogger 5 Junior loggers. The barometric pressure monitored represents both sites 1106 and 1639 due to their close proximity. Data were calibrated over the growing seasons against bi-weekly manual water table level measurements. For later modelling, median groundwater levels between the first of June and the last of August, representing the vegetation period, were used for the available years at each mire.

## Peat elemental concentrations

To describe the variation in nutrient regime between the study sites, we used published surface peat elemental concentrations based on 10 cm deep peat immediately below the water table for sites S10, S18, S26, S52, S56 and S70[58], and reproduced the methodology applied by Wang et al.[58] for sites 1106 and 1639 using the extracted peat cores. Elemental data from the six youngest chronosequence mires emanate from separate peat samples compared to the samples used to calculate MAR, CAR and NAR. Surface peat elemental concentrations of aluminium (Al), calcium (Ca), iron (Fe), potassium (K), magnesium (Mg), manganese (Mn), sodium (Na), phosphorus (P), silicon (Si), sulphur (S) and zinc (Zn) were used, as well as the surface peat pH. Elemental concentrations were based on digestions followed by ICP-OES[15,58].

## Mire age

Analyzes of the temporal variation in MAR, CAR and NAR over the mire chronosequence rely on accurate mire ages ($T_{age}$) estimated from the elevation above sea level ($z$). Land surface age was achieved by combining a $2 \times 2$ m gridded digital elevation model (DEM) with a local shore-displacement curve (Eq. 3[56]). The DEM provided by the Swedish Mapping, Cadastral and Land Registration Authority was based on a LiDAR point cloud with a point density of 0.5-1 points m$^{-2}$, a vertical resolution of 0.3 m and a horizontal resolution of 0.1 m. The shore displacement curve was based on six varved lake sediments in the 29-177 m.a.s.l. elevation range[56].

$$T_{age} = -0.287z^2 + 99.967z \qquad (3)$$

However, it should be noted that mire age may be overestimated using this space-for-time substitution if the mire initiation lags behind land availability. Therefore, the reported age should be considered as the maximum possible mire age.

## Normalized difference vegetation index (NDVI)

Vegetation by the sampling point was described using the normalized difference vegetation index (NDVI[72]) which is a commonly applied vegetation index based on the spectral ratio between the visible red (Red; central wavelength 665 nm) and near-infrared (NIR; central wavelength 842 nm) wavelengths (Eq. 4).

$$NDVI = \frac{NIR - Red}{NIR + Red} \qquad (4)$$

The NDVI was extracted in Google Earth Engine[73] from harmonized atmospherically corrected satellite images from the European Space Agency's Sentinel 2-mission (Level 2 A), with a raster resolution of $10 \times 10$ m. We selected all available Sentinel-2A images with a maximum cloud cover of 30% over the vegetation periods of 2017-2023 and used the most recent NDVI extraction procedures in GEE including masking of clouds and cloud shadows using the Copernicus cloud probability function. Pixels that were defect or covered by shadows, clouds (medium and high probability, as well as cirrus), snow or ice were further masked out before NDVI was calculated. To reduce possible impacts of inter-annual variability in NDVI, we extracted monthly mean NDVI from the pre-vegetation period and green-up in May to post-vegetation period and senescence in September over all available years (2017–2023). To achieve this, we reduced the number of images to one per month, based on the mean value of all remaining pixels containing NDVI data over the 7-year period. Finally, we extracted mean NDVI values and standard deviations for each sampling point.

## Catchment attributes

We defined the catchment area that supports a mire with water and solutes, as all upslope areas that are hydrologically connected to mires through flow paths, governed by surface topography. In this study, we focus on the 'unique' catchment area that a mire does not share with any upslope mires. From the $2 \times 2$ m DEM, we extracted the upslope catchment area for the chronosequence mires using the open-source GIS system Whitebox Geospatial Analysis Tools. Prior to catchment delineation, the DEM was hydrologically pre-processed by burning agricultural streams (defined in the Swedish property map) 1 m into the DEM, carving stream and road intersections into the DEM to allow water to flow through culverts and, finally, we removed sinks from the DEM by applying a breaching algorithm (Lindsay, 2016). We corrected the hydrologically pre-processed DEM for mires by assigning a value of 0 to all mire pixels making them hydrological sinks. From the mire-corrected DEM, we calculated flow direction and flow accumulation using the deterministic eight-direction flow model (D8[74]), and finally delineated catchments, assigning the entire mire area as defined in the Swedish Property Map (Swedish Mapping, Cadastral and Land Registration Authority) as the catchment outlet. Based on this, all upslope flow paths that pass through the mires will be part of the contributing area.

The total mire area and the catchment areas were used to calculate the catchment-to-mire ratio. This ratio was used to estimate the relative importance of catchment support to the mires. Mean catchment elevation, slope, moisture and peat depth were also extracted. Catchment elevation (mean) was extracted directly from the $2 \times 2$ m DEM. This metric is naturally closely related to the estimated mire age and can be used to describe exposure to weathering in upland areas. Catchments at higher elevations have been exposed to the air for a longer period of time compared to catchments at

lower elevations and, consequently, mineral soils in catchments at higher elevations have been exposed to weathering for a longer period of time, resulting in more weathered and potentially more nutrient-poor mineral soils.

From the $2 \times 2$ m DEM, the terrain slope was calculated in SAGA GIS v. 7.9.0 according to Zevenbergen and Thorne[75]. Mean catchment slopes were extracted, with high values representing steep slopes. Mean catchment moisture was calculated from the Swedish soil moisture index ($2 \times 2$ m), which was generated using a machine-learning algorithm based on several terrain indices at different spatial resolutions and ancillary data such as soil properties, runoff and climate data[76]. Moisture data from about 20,000 soil plots distributed across Sweden were used for training and validation. The index describes the likelihood that a pixel is wet (0 = dry, 100 = wet). Further, based on the soil moisture index, the Swedish peat depth map has been generated[77]. The peat map describes the expected peat depth based on the moisture at the site. We expect wetter catchments to cover shallow peat soils[77], while drier catchments may comprise upland soils with shallow humus layers. Taken together, catchment slope, moisture, and peat depth can describe the contact time between water and mineral soil, which may enrich soil water with weathering-derived nutrients. The indices can also describe how easily nutrients and water are transported to mires, or in contrast, remain stuck in stagnant water in the upper catchment area.

### Statistical analyses

As a first data-exploration step, we applied a principal component analysis (PCA) using the multivariate statistical software SIMCA 17, Umetrics, Umeå[78] to compare how MAR, CAR and NAR in lawns and hummocks co-varied with the selected environmental variables. To analyze how the accumulation rates responded to variation in mire age, nutrient regime, and catchment controls we then applied the OPLS (orthogonal projections to latent structures) method. Previous studies have shown that small variation in bedrock composition within the chronosequence area has minor impact on mire nutrient status compared to catchment hydro-topographical properties[15]. Therefore, bedrock properties were not included in the PCA or OPLS models. Using the OPLS approach, variables that co-vary with the determinant can be identified as the model separates variation in the predictive component (x) that is linearly related to the determinant (y) from variation in the predictors (x) that are orthogonal to the determinant (y). Variables that are linearly related to the determinant will have positive or negative loadings on the predictive axis (pq[1]) and will, hence, be more positively or negatively correlated with the determinant the further they are from the origin. Variables that do not correlate with the predictor have high loadings on the orthogonal axis (poso[1]). We used one predictive component and one orthogonal component to describe structures in the dataset and to identify variables that control mire accumulation patterns. Our primary focus is on the predictive component, but for visualization and comparability of the models, we also show one orthogonal component in all models. In SIMCA, we generated OPLS models for MAR, CAR and NAR for lawns and hummocks separately, resulting in six models in total. We defined the respective accumulation rate as determinant (y) and defined all other variables as predictors (x). Before running the models, we applied SIMCA's auto-transform function to identify and transform variables that would approach linearity after log-transformation. Aside from variable loadings, we extracted the variable importance on projection (VIP) based on the predictive component. The VIP identifies variables that are important for the model, such that variables with VIP values >1 are considered as significant[79]. For visualization, we highlighted variables that were significant according to the VIP scores. For these statistical methods that are based on covariation, the chronosequence approach treats the sampled cores as individual data points in the mire population regardless of mire origin.

## Data availability

The $^{210}$Pb data generated in this study, as well as environmental data used as explanatory variables that support the findings of this study have been deposited in Figshare under the accession code https://doi.org/10.6084/m9.figshare.28595561.

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

## Acknowledgements
This study was supported by the Swedish Research Council FORMAS (project 2016-00896 (M.B.N.), 2016-05275 (J.K.), 2018-01217 (C.O.) and 2020-01436 (M.Ö.)). We thank Vildan Toprak and Haijun Peng for their contribution to SCP and 210Pb analyses, as well as Alexandre Dott for assisting with groundwater level monitoring. A special word to Jordi Garcia-Orellana, an exceptional mentor and friend, who recently and unexpectedly left us. Jordi introduced many of us to the use of 210Pb for dating sedimentary deposits. Thanks to his guidance, countless 210Pb profiles have been accurately interpreted and dated, making this and many other studies possible. May he rest in peace.

## Author contributions
B.E., J.L.R., C.O., J.S., K.B., J.K., C.L., C.-M.M., M.B.N. and M.Ö. conceived this study. B.E., J.S., C.-M.M., C.L. and C.O. were involved in data collection. C.O. performed the peat dating, and B.E. performed the data analysis. B.E., with significant support by J.L.R., wrote the original draft. B.E., J.L.R., C.O., K.B., J.K., C.L., M.B.N. and M.Ö. contributed to and discussed the results of this study and improved the original draft. M.B.N., M.Ö., J.K. and C.O. acquired the funding for this work.

## Funding

## Competing interests
The authors declare no competing interests.
