## [Peer Review file · Nature Communications]

Carbon accumulation in recently deposited peat is reduced by increased nutrient supply

Corresponding Author: Dr Betty Ehnavall

Version 0:

Reviewer comments:

Reviewer #1

(Remarks to the Author)

January 2025

Review of "Carbon accumulation in recently deposited peat is reduced by increased nutrient supply" by Ehnavall et al. submitted to Nature Communications

Review conducted by:

Dr Daniel Schillereff

Department of Geography, King's College London

daniel.schillereff@kcl.ac.uk

This study executes a neat real-world experiment using mire chronostratigraphies to establish the influence of groundwater nutrient input on peatland carbon accumulation rates. Their work takes advantage of rapid isostatic uplift in northeastern Sweden to create a type of natural experiment. The work represents an impressive integration of field, lab and computational methods to put forward strong evidence that nutrient input must be considered as a fundamental component of models of peatland development. This supports a growing body of literature pushing peatlands researchers to move (well) beyond classical models of hydroclimate controlling carbon accumulation. I found the majority of the analysis and interpretation to be convincing. I particularly liked Figure 4. Although it's based on relatively few datapoints, the amount of field and lab analysis invested to produce the curve is impressive. I have some moderate areas of concern about the rationale and analysis but I'm fairly confident the authors can address these quite readily. I found the manuscript to be written and presented clearly, although I have a couple of recommendations on re-ordering the structure of the Main results. My comments are listed in the order they appear in the manuscript.

Main comments

Define a mire: I suggest the authors define what they mean by mire and fen from the outset, for the benefit of the broad readership of this journal. I suspect the authors' use of the term 'mire' rather than 'peatland' or 'bog' simply reflects regional differences in language convention rather than any differences in the type of environment being studied. Nevertheless, neither reference 1 nor 2 (Line 35) use the word 'mire', for example, so a reader new to this field may find the mixed terminology quite perplexing. Similarly, clarifying the difference between ombrogenic and minerogenic would be useful.

Justification for the focus on recent peatland dynamics (Lines 71 - 81): I see the logic that understanding contemporary environmental drivers is most pertinent to active acrotelm processes. But this is stated quite briefly. I interpret the authors' concern to be that comparing carbon accumulation rates over the last, say, 5000 years to, for instance, the number of degree days calculated from local weather station or reanalysis data – and thus based on a time series spanning a few years or a few decades is problematic. I don't disagree. But, the way this is expressed is rather vague and rather brief given this is pivotal to the purpose of the study. The authors return to this point on Lines 172 – 173 and indeed this text and references is more convincing than the paragraph in the introduction. I suggest the authors consider some reorganisation so the central narrative is more convincing from the outset.

Rates of carbon passing from acrotelm to catotelm: The authors imply on Lines 76 – 81 that there is a strong relationship

between recent peat accumulation rates and long-term storage in the catotelm. Several papers have explored – and in some cases, questioned – such assumptions on the relationship between acrotelm and catotelm carbon burial rates. I was a bit surprised that this paper doesn't engage more deeply with that literature. For example: Bunsen & Loisel 2020 <https://doi.org/10.1111/gcb.15262>; Young et al. 2021 <https://www.nature.com/articles/s41598-021-88766-8> and indeed work by the second author e.g. Ratcliffe et al. 2017 <https://doi.org/10.1177/0959683617715689>. I'm not suggesting for a moment that the second author has forgotten their papers! But I was surprised there wasn't deeper consideration of this aspect. More detailed commentary in the introduction and probably in the discussion would be valuable, in my view.

Present a more detailed overview of the peatland biogeochemistry: I appreciate the structure of Nature-style papers is to jump from a pithy introduction to the Main results text. However, I found the first few lines (Lines 93 – 99) of Section 2 “Drivers behind recent peat, carbon and nitrogen accumulation rates” to be a rather drastic jump in topic and the ensuing analysis of microtopography and structural models lacks context as a result. In my view, that section should begin by presenting a more detailed overview of the biogeochemistry measurements. More broadly, I would consider reordering the content so that Figure 3 and its associated text and analysis is presented before the ordinations and modelling?

Have the authors duly accounted for differences in catchment geology? The brief summary of regional geology (Lines 215 – 217) suggests some mires may have developed on catchments with somewhat different underlying lithologies. This could alter the chemical composition of inflowing water, and is separate to length of weathering time. I'm a bit surprised geology is not considered as a potential predictor in the OPLS models (Table S2). Are the authors confident catchment geology will have had a negligible role?

Have the authors duly accounted for the role of atmospheric input? I appreciate nutrient and metal deposition on peatland surfaces is likely to be less than mineral input from groundwater in these sorts of minerogenic mires. Nevertheless, presumably atmospheric input will represent a proportionally greater flux to the oldest mires – i.e., those now sitting on the most heavily weathered bedrock. Plus, atmospheric nutrient cycling has increased over the last 1-2 centuries, which could make it a more important factor over the timescale of analysis in this paper. The authors mention rainfall input briefly on Lines 174 – 175, but atmospheric input could be relevant to the discussion e.g. on Lines 130 – 132. Deeper consideration would be welcomed.

Separate the analysis of microtopography from mire-scale dynamics: The text presented on Lines 126 to 132 summarises some of the key analysis and interpretation. The messaging is powerful; I would want the paragraph to finish on this point. The next segment of text (Lines 132 – 137) therefore feels a bit bolted on. Perhaps a separate paragraph or two that focuses specifically on controls on microtopography would be a more effective structure and allow the central messages to shine through a bit more?

Minor comments:

Line 20: are most mires across northerly latitudes minerogenic fens? That feels surprising to me, but

Line 22: Perhaps define 'recent' in the abstract? Readers from varied disciplines might have quite different assumptions.

Line 23: clarify what it is about the input of 'minerogenic water' that results in it having such a profound negative effect on recent carbon accumulation. The sentence in its current form is a little vague – not all readers may immediately grasp the role of nutrient supply.

Lines 98 and 99: A bit unusual to cite Figure 3 immediately after Figure 1 instead of Figure 2. This is related to my comment above about reordering the results section.

Line 147-148: I couldn't follow this sentence, specifically the phrase “a stronger support of weathering derived elements”. Probably just needs rewording slightly.

Reviewer #2

(Remarks to the Author)

Review of “Carbon accumulation in recently deposited peat is reduced by increased nutrient supply”

I have read this manuscript with great interest – partly because I'm quite familiar with the study region. The present manuscript is an output from the “Sävar Rising Coastline Mire Chronosequence project” (<https://slughg.github.io/MiresChrono/about.html>), which take the advantage of using the well-documented and well-established Holocene shoreline land uplift curve for northern Sweden to determine the age of peatlands that are located on different elevations above present-day sea level. This is a neat and highly promising approach, and the “Sävar project” has consequently resulted in a series of publications in a broad range of different journals.

GENERAL COMMENTS

The manuscript is based on the use of modern and “state of the art” analytical peat dating techniques, numerical analyses, digital elevation models, and vegetation indices. Field- and laboratory work is carefully executed, and much effort is devoted to the dating of young surface peat (i.e., < 200-year-old) surface peat) by use of radionuclide Pb-technique, which was combined and supported by an analysis of the vertical distribution of carbonaceous particles origination from the combustion of fossil energy sources. The “dating work” is a strong part of the manuscript because it's generally tricky to determine

stratigraphic age-profiles in young surface peat (i.e., < 200-year-old). In this context I suggest that you should consider a reference to the seminal paper from 1979 by Oldfield et al. in *Oikos* 33: 40-45.

I have, however, a couple of questions and concerns about the manuscript:

1) Peatland location in meter above sea level is not synonymous with peatland age. This because peat initiation (peat accumulation) at given location does not necessary start immediately after land-uplift. A more or less long time-lag is possible. What we need to know is the actual age of the oldest bottom peat at the position wherefrom each peat core was collected. This is important. By knowing the actual age for the position-specific start of peat accumulation, you can also estimate a position-specific “average annual long term C accumulation rate”, which will be of interest as a comparison to your short-term contemporary accumulation rates. In this context it would also be relevant to elaborate some more about long-term differences in peat accumulation among sites. For example, the peat accumulation rate has the potential to be about four times more rapid in mire ID S26 than in mire ID1639 (Supplementary Table S1). What could be the explanation for this anomaly, and may anything of this anomaly be explained by peatland age? Moreover, the position of your core sampling positions in relation to the deepest part of the peatland and peatland margins are also of importance. This needs to be clarified (more about the importance of the within-peatland position of the sample position further below).

2) Peat accumulation rates is to a very large extent driven by contemporary process in the peatland surface (as you state on lines 73-81). It is thus somewhat far-fetched to believe that today’s process in the peatland surface should be related to or impacted by the age of the underlying dead peat. This is obviously the case as clearly shown in this manuscript.

3) Boreal peatlands are characterized by highly variable mass, C, and N accumulation rates in the surface peat, see e.g., Fig. 4 in *Ecology* 79: 2745-2758 and Fig. 2 in *Oikos* 82: 29-36. Please note that I’m not referring to these publications to brag about my own previous work, but to highlight the amount of variability in surface peat accumulation rates. My main reason for doing this is to point at “the single core dilemma”, i.e., how representative are your sampled cores when it comes to the spatial scale of a peatland?

4) Chemical properties of the minerogenic catchment water that enters the peatlands are pointed out as a key-factor as a driver of peat accumulation, which is an interesting main result in this study. Do you have more evidence for this in to support your conclusion? I’m particularly thinking about concentrations of biological important nutrients, minerals etc. Here are not only concentrations of importance, but also flux rates (i.e., amounts of potential nutrients entering the peatland – spring water influence). Furthermore, the within peatland position of the sampled cores are of importance in this context. This because the distance from the peatland margin, where the input of minerogenic water occur, will likely have bearing for the strength of the minerogenic water impact – strongest close to margins, and gradually weaker with increasing distances from the peatland margin.

5) Number of study mires and peat cores are unclear. There are 8 study mires (sites) of different age, but in Figure 3, there are only depicted results from 6 mires (as far as I can see). Kindly explain. In Materials and methods, you mention that 13 cores were oven-dried, and that 4 were freeze-dried: giving a total of 17 cores. However, as I understand your sampling approach (i.e., 2 peat cores from 7 mires, and 1 core from 1 mire), which gives in total 15 cores. Here I may have misunderstood something, but kindly explain.

SPECIFIC AND MINOR COMMENTS + SOME SUGGESTIONS

Line 37: Add a reference after emissions.

Line 39: Consider very (before slow) and add anoxic (after water-saturated).

Lines 40-43: Alternative phrasing suggestion: Importantly, C accumulation over an annual to decadal timescale first occurs by photosynthesis and plant growth in the surface of the biologically highly active surface zone (acrotelm), where decomposition rates are high.....

Lines 45-46: Anoxic is wrong here! Change to more oxygenated conditions in the surface peat.

Line 48: Insert that (before will).

Lines 51-55: About Sphagnum mosses: Consider adding this reference: van Breemen, N. 1995. How Sphagnum bogs down other plants. – *Trends in Ecology and Evolution*, 10: 270–275.

Line 58: Add by (before iron).

Line 61: The concept smaller is vague. I suggest something like: At a local mire spatial scale (or perhaps: At a single mire spatial scale).

Lines 61-70: Here is it relevant to refer to the above-mentioned *Ecology* and *Oikos* papers about surface peat.

Line 70: Consider substituting microforms with hummocks and lawns.

Line 85: Consider substituting rebound with land-uplift.

Lines 134-137: Elaborate by a few more words about how much (i.e., give numbers) CAR differed between hummocks and lawns.

Lines 443 and 447: I guess M.O. should be M.Ö.

Reference list: Carefully check and correct use of capital letters in publication titles.

Kind regards, and best wishes from Mikael Ohlson

Version 1:

Reviewer comments:

Reviewer #1

(Remarks to the Author)
Dear Authors,

Thank you for your considered, comprehensive and convincing responses to my review. Congratulations on putting together such a neat paper.

With best wishes,
Daniel Schillereff

Reviewer #2

(Remarks to the Author)
The authors have done a detailed, very careful, and satisfactory revision of their manuscript. I have no further comments and recommend publication.

Reviewer #1 (Remarks to the Author):

January 2025

Review of “Carbon accumulation in recently deposited peat is reduced by increased nutrient supply” by Ehnvallet al. submitted to Nature Communications

Review conducted by:

Dr Daniel Schillereff

Department of Geography, King’s College London

daniel.schillereff@kcl.ac.uk

‘This study executes a neat real-world experiment using mire chronostratigraphies to establish the influence of groundwater nutrient input on peatland carbon accumulation rates. Their work takes advantage of rapid isostatic uplift in northeastern Sweden to create a type of natural experiment. The work represents an impressive integration of field, lab and computational methods to put forward strong evidence that nutrient input must be considered as a fundamental component of models of peatland development. This supports a growing body of literature pushing peatlands researchers to move (well) beyond classical models of hydroclimate controlling carbon accumulation. I found the majority of the analysis and interpretation to be convincing. I particularly liked Figure 4. Although it’s based on relatively few datapoints, the amount of field and lab analysis invested to produce the curve is impressive. I have some moderate areas of concern about the rationale and analysis but I’m fairly confident the authors can address these quite readily. I found the manuscript to be written and presented clearly, although I have a couple of recommendations on re-ordering the structure of the Main results. My comments are listed in the order they appear in the manuscript.’

We are delighted to hear that the reviewer appreciates our study approach that combines field, lab and computational methods to support the growing body of literature challenging traditional theories on peatland development.

Main comments

‘Define a mire: I suggest the authors define what they mean by mire and fen from the outset, for the benefit of the broad readership of this journal. I suspect the authors’ use of the term ‘mire’ rather than ‘peatland’ or ‘bog’ simply reflects regional differences in language convention rather than any differences in the type of environment being studied. Nevertheless, neither reference 1 nor 2 (Line 35) use the word ‘mire’, for example, so a reader new to this field may find the mixed terminology quite perplexing. Similarly, clarifying the difference between ombrogenic and minerogenic would be useful.’

We agree that the word ‘mire’ is partly regional in origin, as the term is widely used in Fennoscandia but less so, for example, in North America. However, there is a hydrological and ecological distinction between the different terms. We prefer to use the term ‘mire’ as it covers both minerogenic peatland ecosystems (i.e. catchment supported fens) and ombrogenic peatland ecosystems (i.e. purely rainwater fed bogs). In this way, the term ‘mire’ reflects the peat-forming ecosystem, in contrast to the term ‘peatland’, which can also refer to peat soils that do not have typical peat-forming vegetation and are, therefore, not considered as ‘mires’ from an ecosystem functioning perspective. This distinction is important to keep in mind, for example, in heavily drained peatland forests where the original peat-forming mire vegetation may have been replaced by typical forest vegetation, resulting in a peat soil with typical forest vegetation.

Action: We have added the following explanations to the main text of the manuscript, but have not included explanations in the abstract due to word limits and a desire to keep the abstract focused on the core message:

Line 35: *'mires, i.e. peat-forming wetland ecosystems'*

Line 58: *'In minerogenic, i.e. groundwater-fed mires (fens)'*

Lines 243-244: *'ombrogenic mires (bogs), which receive water and nutrients only from precipitation'*

'Justification for the focus on recent peatland dynamics (Lines 71 - 81): I see the logic that understanding contemporary environmental drivers is most pertinent to active acrotelm processes. But this is stated quite briefly. I interpret the authors' concern to be that comparing carbon accumulation rates over the last, say, 5000 years to, for instance, the number of degree days calculated from local weather station or reanalysis data – and thus based on a time series spanning a few years or a few decades is problematic. I don't disagree. But, the way this is expressed is rather vague and rather brief given this is pivotal to the purpose of the study. The authors return to this point on Lines 172 – 173 and indeed this text and references is more convincing than the paragraph in the introduction. I suggest the authors consider some reorganisation so the central narrative is more convincing from the outset.'

We focus on contemporary carbon accumulation rates because these are controlled by present-day environmental conditions which can be monitored. Additionally, these rates are crucial for the transfer of peat from the acrotelm to the catotelm, which enables long-term peat accumulation.

Action: According to the reviewer's comment, we have rewritten the following paragraph:

Lines 76-93. 'When considering specifically the processes leading to long-term C accumulation, it is crucial to focus on the oxic acrotelm where most of the primary production and decay occur. It is well known that near-surface C accumulation rates are higher in comparison with in deeper peat layers as peat becomes progressively older and thus, more degraded with depth^{27,28}. However, these accumulation rates can also represent varying productivity and decay dynamics, with different implications for the C input into the relatively stable, anoxic catotelm²⁹. For instance, increased decay and productivity may increase long-term C inputs in some sites while reducing them in others³⁰. Therefore, interpreting C accumulation rates requires careful consideration taking into account both organic matter decay rates and the acrotelm residence time. By examining the acrotelm, we can better relate present environmental conditions, including morphological, geochemical and climatic conditions, to contemporary surface peat rather than peat formed thousands of years ago³¹. Hence, understanding the drivers of recent peat accumulation rates is key as they have major implications for the contemporary C balance of high-latitude mires and the C currently entering or soon to enter long-term storage in the catotelm.'

'Rates of carbon passing from acrotelm to catotelm: The authors imply on Lines 76 – 81 that there is a strong relationship between recent peat accumulation rates and long-term storage in the catotelm. Several papers have explored – and in some cases, questioned – such assumptions on the relationship between acrotelm and catotelm carbon burial rates. I was a bit surprised that this paper doesn't engage more deeply with that literature. For example: Bunsen & Loisel 2020 <https://doi.org/10.1111/gcb.15262>; Young et al. 2021 <https://www.nature.com/articles/s41598-021-88766-8> and indeed work by the second author e.g. Ratcliffe et al. 2017 <https://doi.org/10.1177/0959683617715689>. I'm not suggesting for a moment that the second author has forgotten their papers! But I was surprised there wasn't deeper consideration of this aspect. More detailed commentary in the introduction and probably in the discussion would be valuable, in my view.'

We appreciate the reviewer's feedback and would like to clarify that we do not wish to imply a strong relationship between recent and long-term peat accumulation rates. Rather, we aim to highlight that the surface peat layers are the most important for studying productivity and decay processes, and that these processes can, and in our case, do affect the long-term C inputs.

Action: In the introduction (Line 76-93, as shown in the previous reply), we discuss the considerations that should be made when working with recent accumulation rates, and later in the discussion (line 187-202) we provide information about the C dynamics in the acrotelm, specifically the proportion of primary production remaining after 100 years, which is well below the water table depth in all our cores. We also describe the sensitivity to decay rates of the magnitude we report, e.g. 0.01-0.02 yr⁻¹ with the result that C remaining after 100 years would be on average 52% less with a doubling of decay.

Lines 186-200: 'Interpreting recent, i.e. near surface acrotelm, MAR and CAR requires special consideration with regards to whether the dynamics we record here will persist to the long-term C storage pool in the catotelm peat²⁸. For instance, Ohlson and Økland²⁰ showed that the top 20 cm of peat could represent anywhere from 7 to 173 years of peat accumulation. It follows that sites with a short acrotelm residence time of less than or a few decades will be relatively insensitive to the rate of decomposition, in contrast to sites with a long acrotelm residence time of up to a decade, where aerobic decomposition has more time to act and therefore more influence, leading to greater C losses²⁹.

At our sites, peat from ~1920 was found between 29 and 47 cm depth, well below the water table in all cores and representing the catotelm. After 100 years of decay, an average of 41% of productivity, was found to remain (range: 14 % to 89 %, data in repository), indicating active peat accumulation in the anoxic catotelm at a mean rate of ~ 50 g m⁻² yr⁻¹. Doubling the decay rate across all sites, which is comparable to the difference in decay rates between the lawns in the oldest and youngest mires, resulted in an average 52% reduction in C remaining after 100 years, with individual cores showing reductions between 11% and 86%. These variations in recent MAR, CAR and decay rates are therefore expected to drive changes in the long-term C storage of the catotelm.'

Additionally, we have incorporated the suggested literature as references:

Bunsen & Loisel 2020 <https://doi.org/10.1111/gcb.15262>

Young et al. 2021 <https://www.nature.com/articles/s41598-021-88766-8>

Ratcliffe et al. 2017 <https://doi.org/10.1177/0959683617715689>.

'Present a more detailed overview of the peatland biogeochemistry: I appreciate the structure of Nature-style papers is to jump from a pithy introduction to the Main results text. However, I found the first few lines (Lines 93 – 99) of Section 2 "Drivers behind recent peat, carbon and nitrogen accumulation rates" to be a rather drastic jump in topic and the ensuing analysis of microtopography and structural models lacks context as a result. In my view, that section should begin by presenting a more detailed overview of the biogeochemistry measurements. More broadly, I would consider reordering the content so that Figure 3 and its associated text and analysis is presented before the ordinations and modelling?'

We acknowledge that the format of this Nature-style paper makes the story somewhat compressed. However, we hope you find that the revised version has now been improved.

Action: To provide background to our analyses, and better context for the discussed peatlands, we have:

- Added a brief description of the biogeochemistry measurements section 2.
- Restructured the entire section 2 according to the reviewer's suggestions.

- Reordered the figures to ensure a more logical flow:

Lines 105-228: *'We studied recent accumulation rates of peat mass (MAR: 170-280 g m⁻² yr⁻¹), carbon (CAR: 85-140 g C m⁻² yr⁻¹) and nitrogen (NAR: 1.1-2.0 g N m⁻² yr⁻¹) over the past century using ²¹⁰Pb chronologies. Over the entire mire population, CAR was up to 75 g C m⁻² yr⁻¹ higher in lawns than in hummocks. To identify the drivers behind changes in accumulation rates, we evaluated various mire and catchment properties as explanatory variables. Average long-term peat accumulation rates based on the total peat depth would be difficult to interpret due to oligotrophication and/or ombrotrophication, with subsequent changes in peat accumulation rates, that occur across the age gradient. Total peat depth also does not necessarily reflect the rate of peat accumulation, as peat porosity varies between sites, usually with more porous peat in the younger mires. Therefore, we expect that recent accumulation rates will better reflect differences in peat accumulation across the trophic gradient than long-term average accumulation rates.'*

Neither mire age, peat depth, nor water table depth alone could directly explain changes in peat accumulation rates (Fig.1), suggesting that ecosystem ageing per se is not the primary driver. However, in hummocks, accumulation rates were indirectly influenced by mire age and the build-up of the hummock microform, as evidenced by the strong contribution of age-related variables (age, catchment elevation, peat depth, mire area, catchment area) to the orthogonal projections to latent structures model (OPLS). In contrast, mire age did not affect accumulation rates in the more spatially dominant lawns (Fig.2). Furthermore, mire age and water table depth were not correlated, indicating that the observed temporal patterns associated with mire age cannot be attributed solely to differences in water table levels or acrotelm depth (Fig.3). This suggests that additional factors must also play a role in shaping the dynamics peat accumulation.

We propose that nutrient availability from weathering processes enhance peat decay in the youngest minerogenic mires studied, resulting in a decreasing trend in lawn decay rates (based on Clymo's²⁹ model) with increasing mire age (Fig.4a, R² = 0.9, p = 0.01). Elements in peat can promote peat decay both directly, by participating in electron transfer, and indirectly as nutrients influencing plant community composition. Higher nutrient support from the catchment in the studied mires was reflected in higher productivity in hummocks and higher vegetation greenness (Fig.3), in agreement with previous studies in the area¹⁵.

Mineral soils in catchments closer to the present coastline of the isostatically rising land have been exposed to weathering for a shorter period of time compared to soils further inland³². This has profound implications for the composition and flux of nutrients exported to mires across the age gradient¹⁵. As the landscape ages, lateral inflow of nutrients to mires declines, and vertical nutrient inputs decrease with increasing peat depth³³. Consequently, peat-forming vegetation in older mires must rely more on recycled nutrients, which are depleted over time as nutrients become buried and immobilized in partially decomposed organic matter³⁴. Atmospheric deposition may act as a background source of nutrients, in addition to inputs from soil or water. Given the proximity of the youngest and oldest mire (< 20 km), deposition rates are likely to be relatively uniform across the chronosequence area, playing potentially a greater role in older mires compared to the younger mires, which are also fed by weathering products¹⁵. Interestingly, N deposition can potentially promote peat decay in shrub-covered mires, with consequent peat C losses, due to decreased ericoid mycorrhizal activity in peatlands³⁵.

Interpreting recent, i.e. acrotelm, MAR and CAR requires special consideration with regards to whether the dynamics we record here will persist to the long-term C storage pool in the catotelm peat²⁸. For instance, Ohlson and Økland²⁰ showed that the top 20 cm of peat could represent anywhere from 7 to 173 years of peat accumulation. It follows that sites with a short acrotelm residence time of less than or a few decades will be relatively insensitive to the rate of decomposition, in contrast to sites with a long acrotelm residence time of up to a century, where

aerobic decomposition has more time to act and therefore more influence, leading to greater C losses²⁹.

At our sites, peat from ~1920 was found between 29 and 47 cm depth, well below the average water table in all cores and representing the catotelm. After 100 years of decay, an average of 41% of productivity, was found to remain (range: 14 % to 89 %, data in repository), indicating active peat accumulation in the anoxic catotelm at a mean rate of ~ 50 g m⁻² yr⁻¹. Doubling the decay rate across all sites, which is comparable to the difference in decay rates between the lawns in the oldest and youngest mires, resulted in an average 52% reduction in C remaining after 100 years, with individual cores showing reductions between 11% and 86%. These variations in recent MAR, CAR and decay rates are therefore expected to drive changes in the long-term C storage of the catotelm.

MAR and CAR have earlier been proposed to be negatively affected by variables associated with minerogenic water inputs, such as surface peat concentrations of Si, Fe and Mg³⁶. Concentrations of these elements co-varied with the catchment-to-mire ratio (C-to-M; Fig.3), suggesting that large catchments contribute with more weathering-derived elements to recipient mires. Our results indicate that hummocks are more sensitive to nutrient availability, mire size and ageing compared to lawns whereas lawns are more influenced by catchment size (Fig.2).

Based on our OPLS models, the dynamics of lawns follow our hypothesis: higher decay rates covary with lower MAR, CAR, NAR, while higher primary productivity covaries with higher accumulation rates. Looking at productivity and decay rates separately (Fig.4), it is clear that lawn decay rates decline as mires ages, likely due to reduced minerogenic nutrient inputs (Fig.4a, R² = 0.9, p = 0.01). In contrast, lawn primary productivity remain constant over time (around 270 g m² yr⁻¹), with no clear dependence on mire age or water table depth (Fig.4). This suggests that nutrient availability alone is not the main driver of lawn peat productivity. Because of the high productivity rate compared to total mass loss, nutrient conditions have not significantly altered MAR in lawns over the past 100-150 years (Fig.2). In addition, the residence time of peat in the acrotelm is likely to be shorter in lawns than in hummocks due to a generally shallower water table, which may further limit the impact of peat acrotelm decay in lawns compared to hummocks.'

‘Have the authors duly accounted for differences in catchment geology? The brief summary of regional geology (Lines 215 – 217) suggests some mires may have developed on catchments with somewhat different underlying lithologies. This could alter the chemical composition of inflowing water, and is separate to length of weathering time. I’m a bit surprised geology is not considered as a potential predictor in the OPLS models (Table S2). Are the authors confident catchment geology will have had a negligible role?’

Despite some differences in the lithology, as mentioned in the methods, previous studies from the area (e.g. Ehnvall et al., 2023: <https://doi.org/10.1016/j.scitotenv.2023.165132>) have shown that the role of catchment geology is of minor importance for peat and mire plant chemical composition compared to mire age and topographic position. Instead, weathering drives the primary gradient of incoming weathering products across the chronosequence, with phosphorus, for example, being released through appetite weathering within only a few hundred years (Giesler, 2010). This weathering gradient is reflected in both the peat elemental concentrations and mire age, both of which are both included in the OPLS model. Based on this, we did not include measures of catchment geology in the OPLS model for this study. Importantly, Ehnvall et al. (2023) analyzed a partly different selection of chronosequence mires than the present study, though covering the same age range. The main discrepancies in soil properties across the chronosequence relate to floodplains along the Sävar river, which cut through the chronosequence. However, no floodplain mires were included in either the present, or the Ehnvall et al. (2023) paper.

Giesler, 2010. *Rapid transformation of P across a podzol chronosequence in Northern Sweden. Geochim. Cosmochim. Acta 74, A329.*

Action: To clarify why catchment geology was not included in the analyses we added the following:

Lines 500-503: *‘Previous studies have shown that small variation in bedrock composition within the chronosequence area has minor impact on mire nutrient status compared to catchment hydro-topographical properties¹⁵. Therefore, bedrock properties were not included in the PCA or OPLS models.’*

‘Have the authors duly accounted for the role of atmospheric input? I appreciate nutrient and metal deposition on peatland surfaces is likely to be less than mineral input from groundwater in these sorts of minerogenic mires. Nevertheless, presumably atmospheric input will represent a proportionally greater flux to the oldest mires – i.e., those now sitting on the most heavily weathered bedrock. Plus, atmospheric nutrient cycling has increased over the last 1-2 centuries, which could make it a more important factor over the timescale of analysis in this paper. The authors mention rainfall input briefly on Lines 174 – 175, but atmospheric input could be relevant to the discussion e.g. on Lines 130 – 132. Deeper consideration would be welcomed.’

Atmospheric phosphorus inputs in northern Sweden are generally low today, with less than 75 g ha⁻¹ yr⁻¹ of phosphorus (Karlsson et al., 2021). Historically, this deposition was even lower. Similarly, the deposition of nitrogen and sulphur in northern Sweden is low today compared to a century ago, with nitrogen deposition today as low as 2 kg ha⁻¹ yr⁻¹ (Karlsson et al., 2024). It is important to note that the chronosequence spans a relatively constrained area with only ca 20 km between the youngest and the oldest sites. Hence, atmospheric inputs should be relatively uniform across the chronosequence area and constitute ‘background levels’ of nutrient supply to the mire vegetation. These ‘background levels’ of nutrients could have a more important role in nursing the mire vegetation in older mires, as the reviewer suggests, but the overall contribution of atmospherically derived nutrients is likely to be negligible due to the very low deposition rates.

Karlsson, P.E., Pihl Karlsson, G., Hellsten, S., 2021. *Deposition av fosfor till skog och öppen mark I Sverige. SMED Rapport Nr 25 2021. (Abstract available in English).*

Pihl Karlsson, G., Akselsson, C., Hellsten, S., Karlsson, P.E., 2024. *Atmospheric deposition and soil water chemistry in Swedish forests since 1985 – Effects of reduced emissions of sulphur and nitrogen. Science of the total environment 25, 913. <https://doi.org/10.1016/j.scitotenv.2023.169734>*

Action: To address the issue of possible atmospheric input of nutrients, we have added the following (as already shown as part of the reorganization of the results part):

Lines 175-179: *‘Atmospheric deposition may act as a background source of nutrients, in addition to inputs from soil or water. Given the proximity of the youngest and oldest mire (< 20 km), deposition rates are likely to be relatively uniform across the chronosequence area, playing potentially a greater role in older mires compared to the younger mires, which are also fed by weathering products¹⁵.’*

‘Separate the analysis of microtopography from mire-scale dynamics: The text presented on Lines 126 to 132 summarises some of the key analysis and interpretation. The messaging is powerful; I would want the paragraph to finish on this point. The next segment of text (Lines 132 – 137) therefore feels a bit bolted on. Perhaps a separate paragraph or two that focuses specifically on controls on microtopography would be a more effective structure and allow the central messages to shine through a bit more?’

Thank you for these suggestions. Unfortunately, the line numbering seems to be incorrect, but based on the description of the paragraph content, we hope that we interpret the suggestions correctly. To make

the message stronger, we solved this comment when we restructured the ‘results part’, as shown on page 4 of this document.

Minor comments:

Line 20: are most mires across northerly latitudes minerogenic fens? That feels surprising to me, but

At northern latitudes, ombrogenic conditions rarely develop, as the peat surface cannot rise above its surroundings due to unfavorable climatic conditions. It is important to note that our classification of fens is based purely on hydrology. While the vegetation can still be typical ‘bog vegetation’ (e.g. *Sphagnum* dominated), these systems remain oligotrophic, minerogenic mires (i.e. nutrient-poor fens) from a hydrological/biogeochemical perspective.

No action was taken.

Line 22: Perhaps define ‘recent’ in the abstract? Readers from varied disciplines might have quite different assumptions.

Action: We have added a definition of ‘recent’.

Lines 21-23: *‘Here we show that for boreal fens in our mire chronosequence study, the main negative influence on recent (i.e. past century) carbon accumulation is...’*

Line 23: clarify what it is about the input of ‘minerogenic water’ that results in it having such a profound negative effect on recent carbon accumulation. The sentence in its current form is a little vague – not all readers may immediately grasp the role of nutrient supply.

Action: We have added a clarification to the role of minerogenic water as a source of nutrients.

Lines 21-24: *‘Here we show that for boreal fens in our mire chronosequence study, the main negative influence on recent (i.e. past century) carbon accumulation is the input of minerogenic water. This effect on carbon accumulation from nutrient containing water is most pronounced in younger mires.’*

Lines 98 and 99: A bit unusual to cite Figure 3 immediately after Figure 1 instead of Figure 2. This is related to my comment above about reordering the results section.

Action: The reordering of the results section, as shown earlier in this document, has resolved the issue of the unusual citation.

Line 147-148: I couldn’t follow this sentence, specifically the phrase “a stronger support of weathering derived elements”. Probably just needs rewording slightly.

Action: We have rephrased this to make a clearer message:

Lines 210-213: *‘Concentrations of these elements co-varied with the catchment-to-mire ratio (C-to-M; Fig.3), suggesting that large catchments contribute with more weathering-derived elements to recipient mires.’*

Reviewer #2 (Remarks to the Author):

Review of “Carbon accumulation in recently deposited peat is reduced by increased nutrient supply”

I have read this manuscript with great interest – partly because I’m quite familiar with the study region. The present manuscript is an output from the “Sävar Rising Coastline Mire Chronosequence project” (<https://slughg.github.io/MiresChrono/about.html>), which take the advantage of using the well-documented and well-established Holocene shoreline land uplift curve for northern Sweden to determine the age of peatlands that are located on different elevations above present-day sea level. This is a neat and highly promising approach, and the “Sävar project” has consequently resulted in a series of publications in a broad range of different journals.

We are pleased that Reviewer #2 also appreciates the study approach and that he is familiar with previous projects and publications from the chronosequence area.

GENERAL COMMENTS

The manuscript is based on the use of modern and “state of the art” analytical peat dating techniques, numerical analyses, digital elevation models, and vegetation indices. Field- and laboratory work is carefully executed, and much effort is devoted to the dating of young surface peat (i.e., < 200-year-old) surface peat) by use of radionuclide Pb-technique, which was combined and supported by an analysis of the vertical distribution of carbonaceous particles origination from the combustion of fossil energy sources. The “dating work” is a strong part of the manuscript because it’s generally tricky to determine stratigraphic age-profiles in young surface peat (i.e., < 200-year-old). In this context I suggest that you should consider a reference to the seminal paper from 1979 by Oldfield et al. in *Oikos* 33: 40-45.

We are very grateful that the reviewer recognized the significant amount of work involved in dating the peat cores. Normally, the methodology used for dating peat cores is briefly included as part of the methods section, and the effort involved in terms of laboratory work and data interpretation is not always visible. We fully agree that the study by Oldfield et al. 1979 is a clear example on how a ^{210}Pb concentration profile in peat cores needs to be interpreted carefully and together with independent markers to be able to provide accurate peat ages and accumulation rates.

Action: In the revised version of the manuscript, we have included Oldfields’ work to highlight the need to use independent dating techniques, as well as the challenges of applying ^{210}Pb dating in mires with contrasting microtopography (i.e. hummocks, hollows, lawns), peat humification, and hydrology. We also included Olid et al., 2016 as a reference:

Lines 334-338: ‘On estimating accurate ages due to the potential downward mobility of ^{210}Pb in the uppermost peat layers⁶²⁻⁶³, profile chronologies were determined based on ^{210}Pb using the Constant Flux: Constant Sedimentation (CF:CS) model⁶⁴ or the Constant Rate of Supply (CRS) model, depending on which model provided an accumulation rate that was more consistent with the SCP record⁶⁵.’

Lines 370-372: ‘However, we found some difficulties when applying it to the hummocks, likely due to the nonhomogeneous nature of the peat and the strong influence of water table fluctuations, which might have affected the retention of ^{210}Pb and introduce uncertainties in the derived ages⁶².’

Lines 347-348: ‘The ^{210}Pb profiles showed significant variations from the expected exponential decay, requiring an independent chronological marker to validate the ^{210}Pb -derived chronology⁶².’

*Oldfield, F., Appleby, R.S., Cambray, J.D., Eakins, J.D., Barber, K.E., Battarbee, R.W., Pearson, G.R., Williams, J.M., 1979. ^{210}Pb , ^{137}Cs , and ^{239}Pu profiles in ombrotrophic peat. *Oikos* 33: 40-45.*

Olid, C., Diego, D., Garcia-Orellana, J., Cortizas, A.M., Klaminder, J., 2016. Modeling the downward transport of ²¹⁰Pb in Peatlands: Initial Penetration-Constant Rate of Supply (IP-CRS) model. Sci.Total Environ. 541, 1222-1231.

I have, however, a couple of questions and concerns about the manuscript:

1) Peatland location in meter above sea level is not synonymous with peatland age. This because peat initiation (peat accumulation) at given location does not necessary start immediately after land-uplift. A more or less long time-lag is possible. What we need to know is the actual age of the oldest bottom peat at the position wherefrom each peat core was collected. This is important. By knowing the actual age for the position-specific start of peat accumulation, you can also estimate a position-specific “average annual long term C accumulation rate”, which will be of interest as a comparison to your short-term contemporary accumulation rates. In this context it would also be relevant to elaborate some more about long-term differences in peat accumulation among sites. For example, the peat accumulation rate has the potential to be about four times more rapid in mire ID S26 than in mire ID1639 (Supplementary Table S1). What could be the explanation for this anomaly, and may anything of this anomaly be explained by peatland age? Moreover, the position of your core sampling positions in relation to the deepest part of the peatland and peatland margins are also of importance. This needs to be clarified (more about the importance of the within-peatland position of the sample position further below).

We agree that peat initiation, leading to peat accumulation and eventually the establishment of a mire ecosystem, may lag behind land availability in these postglacial landscapes. However, the date of coastline emergence has been uncontroversially used as a proxy date of mire initiation in several highly cited papers, such as Clymo et al., (1998) and Tolonen and Turunen (1996), which together have more than 900 citations. These two studies were based on mires along the Eastern side of the Bothnian Bay, with a land surface subject to similar environmental conditions as our Swedish study mires. Although we do not have access to bottom peat dates from each of the peat core sites collected in the present study, we would like to draw your attention to our paper by Ehnvall et al. (2023) (<https://doi.org/10.1016/j.quascirev.2023.107961>), which included a figure (Fig A.2 below) in the supporting information that increases the reliability and credibility of our interpretations. The figure includes mires with available peat dates from the wider area (Västerbotten and Norrbotten), and it supports the chronosequence approach in this Swedish post-glacial coastal landscape with strong isostatic rebound.

Clymo, R.S., Turunen, J., Tolonen, K., 1998. Carbon Accumulation in Peatland. Oikos 81, 368. <https://doi.org/10.2307/3547057>

Tolonen, K., Turunen, J., 1996. Accumulation rates of carbon in mires in Finland and implications for climate change. The Holocene 6, 171–178. <https://doi.org/10.1177/095968369600600204>

Fig. A.2. Comparison of land surface ages and calibrated ^{14}C ages (Klarqvist et al., 2001) for three mires from different chronosequences in the study area: Stor-Ämyran, Sjulsmýran and Rismýran from the Hörnefors, Sävar and Kalix chronosequences. Elevation based mire ages are linearly related to ^{14}C ages ($R^2 = 0.98$), although, in all mires the modelled mires ages are overestimating the actual ages (cal BP) with several hundreds of years.

The main deviation from the expected age gradient would be due to different mire development pathways. If a mire is initiated through terrestrialization rather than the dominating primary mire formation, mire initiation may be more delayed relative to land exposure. In the context of a chronosequence study, however, the reported age should be considered as the maximum possible age of the ecosystem. When interpreted in this way, any potential delay in peat initiation becomes less significant.

Action: To clarify that the land-surface ages may overestimate the actual mire age, we have added the following:

Lines 435-436: *'Mire age may be overestimated using this space-for-time substitution if the mire initiation lags behind land availability. Therefore, the reported age should be considered as the maximum possible mire age.'*

We appreciate the suggestion to include estimates of the average annual long-term C accumulation rate as a comparison between the short-term and long-term accumulation rates. However, while we do not have information on the exact timing of peat initiation (which is why we use the chronosequence approach) and, hence, cannot provide such a figure, we also prefer to keep the manuscript focused on peat accumulation rates and the contemporary drivers behind them, rather than adding a thorough comparison between short and long-term accumulation rates. In the context of this mire chronosequence, average peat accumulation rates would be difficult to interpret because of the oligotrophication and/or ombrotrophication and changes in peat accumulation rates that occur across the chronosequence. Average long-term accumulation rates would not reflect these changes. Hence, we expect that recent accumulation rates better reflect the differences in peat accumulation across the trophic gradient than long-term average accumulation rates. We discuss the importance of the core sampling position further down in this document, along with a comment on variable mass, C and N accumulation rates.

The potential for up to four times higher long-term peat accumulation in the intermediate aged S26 compared to the old 1639 mentioned by the reviewer, seem to be based on the total peat depth by the sampling location and the age of the mires based on the land surface age. It is important to note, that the total peat depth does not necessarily reflect the rate of peat accumulation, as the porosity of the peat varies between sites, usually with more porous (i.e. relatively deeper) peat in the younger mires. I have demonstrated this in my doctoral thesis, based on the chronosequence mires used in this study and for a larger selection of chronosequence mires in the Sävar area (Figure 11; Ehnvall, 2023). The figure

illustrates a rapid decrease in the ‘peat growth rate’, due to reduced peat porosity. It shows that there is some variation in the peat growth rate, but also that the mires used in this study (Figure 11c) follow the expected decrease in peat growth.

Figure 11. a) Depths by mire margins (N = 160, blue), as well as intermediate (N = 160, green) and central (N = 80, yellow) positions over the past 6 000 years in mires visited for paper I. b) Average peat height growth rates since peat initiation estimated from peat depth, and mire bottom age estimated from the elevation above sea level and a local shore displacement curve (Renberg and Segerström, 1981) over the past 6 000 years. c) Peat height growth rates over the past 8 000 years for mires included in paper II.

Ehnavall, B., 2023. *Catchment controls on mire properties in the post-glacial landscape (Doctoral thesis)*. Acta Universitatis Agriculturae Sueciae, 73. <https://doi.org/10.54612/a.2hq3ebpddu>

Action: We have added these sentences to reflect shifts in accumulation rates across the studied mires, which the chronosequence approach predicts, and to elaborate on the long-term differences in peat accumulation:

Lines 109-115: ‘Average long-term peat accumulation rates based on the total peat depth would be difficult to interpret due to oligotrophication and/or ombrotrophication, with subsequent changes in peat accumulation rates, that occur across the age gradient. Total peat depth also does not necessarily reflect the rate of peat accumulation, as peat porosity varies between sites, usually with more porous (i.e. relatively deeper) peat in the younger mires. Therefore, we expect that recent accumulation rates based on ²¹⁰Pb chronologies will better reflect differences in peat accumulation across the trophic gradient than long-term average accumulation rates.’

2) Peat accumulation rates is to a very large extent driven by contemporary process in the peatland surface (as you state on lines 73-81). It is thus somewhat far-fetched to believe that today’s process in the peatland surface should be related to or impacted by the age of the underlying dead peat. This is obviously the case as clearly shown in this manuscript.

Surface processes are indeed important for the burial of freshly produced peat in the water saturated catotelm, as we discuss in the manuscript. The importance of age that we discuss is not primarily related

to the age of the underlying peat, but to the age of the catchment, i.e. the time since land exposure from the sea, with subsequent peat initiation and mire establishment. In this way, the age influences the weathering of mineral soils in the surrounding catchment area, and the nutrient input to the mire. No action was taken.

3) Boreal peatlands are characterized by highly variable mass, C, and N accumulation rates in the surface peat, see e.g., Fig. 4 in *Ecology* 79: 2745-2758 and Fig. 2 in *Oikos* 82: 29-36. Please note that I'm not referring to these publications to brag about my own previous work, but to highlight the amount of variability in surface peat accumulation rates. My main reason for doing this is to point at "the single core dilemma", i.e., how representative are your sampled cores when it comes to the spatial scale of a peatland?

This part of comment 1) is also handled in the reply below: 'Moreover, the position of your core sampling positions in relation to the deepest part of the peatland and peatland margins are also of importance. This needs to be clarified (more about the importance of the within-peatland position of the sample position further below).'

Peatland C accumulation dynamics can indeed be highly variable, but in the case of our cores these were remarkably uniform, at least in recent, i.e. 100-year accumulation rates, which averaged 3.6 mm yr⁻¹ with a standard deviation of only 0.7 mm yr⁻¹. In our study design we focused only on more central, deeper areas of peat away from the margins. This makes it all more remarkable that we see the influence of the catchment area on the decay rates of the surface peat. From the two oldest mires (ID 1639 and 1106), these cores were collected from the center of the mire along the sampling transects used in Ehnvall et al. (2023; <https://doi.org/10.1016/j.scitotenv.2023.165132>). We are confident that these cores represent a deep section of the mire, relatively unaffected by edge effects. It has also been demonstrated how peat depth varies along edge-to-edge transects in mires with different sized mires within the chronosequence area (Ehnvall, 2023; Figure 16). Although peat depth may vary depending on local topography/bathymetry, these findings support the choice of sampling from the mire center. The remaining cores were also sampled from the mire expanse but not necessarily from the center. Instead, the cores were collected from the same sampling points used in previous studies (Wang et al. (2021) and Smeds et al. (2022)).

Smeds, J., Öquist, M., Nilsson, M. B. & Bishop, K. A Simplified Drying Procedure for Analysing Hg Concentrations. *Water, Air, Soil Pollut.* **233**, 216 (2022)

Wang, B. et al. Biogeochemical influences on net methylmercury formation proxies along a peatland chronosequence. *Geochim. Cosmochim. Acta* **308**, 188–203 (2021).

Figure 16. Mire depth profiles across all mire transects sampled as a part of fieldwork related to paper I.

Ehnvall, B., 2023. Catchment controls on mire properties in the post-glacial landscape (Doctoral thesis). *Acta Universitatis Agriculturae Sueciae*, 73. <https://doi.org/10.54612/a.2hq3ebpddu>

Action: To specify where along the mire margin-expense continuum the cores were collected, we have added the following:

Lines 310-313: *'Peat cores from the two oldest mires (ID 1106 and 1639) were sampled from the centre of the mire according to Ehnvall et al¹⁵, while cores from the remaining mires (ID S10, S18, S26, S52, S65 and S70) were collected from locations previously used in studies such as Wang et al⁵⁸.'*

Although we do not include multiple cores from each mire, we have sampled uniform microtopographies (lawns and hummocks separately) and pooled these results across the age gradient. This sampling strategy overcomes the one-core-dilemma, as data points from the entire mire population are used in the statistical analyses. For example, in Figure 4, the cores from the individual mires constitute a sample population on which the regression is based. We would also like to draw back your attention to the initial comment by Reviewer #1 related to the time and effort involved in generating each data point. At some point we simply cannot add more replicates, but have to choose between studying one or a few mires in depth, or applying the chronosequence approach based on several mires in a mire population.

Action: To emphasize the importance of the chronosequence approach in replicating data across the mire population, we have added the following:

Lines 520-522: *'For these statistical methods that are based on covariation, the chronosequence approach treats the sampled cores as individual data points in the mire population regardless of mire origin.'*

4) Chemical properties of the minerogenic catchment water that enters the peatlands are pointed out as a key-factor as a driver of peat accumulation, which is an interesting main result in this study. Do you have more evidence for this in to support your conclusion? I'm particularly thinking about concentrations of biological important nutrients, minerals etc. Here are not only concentrations of importance, but also flux rates (i.e., amounts of potential nutrients entering the peatland – spring water influence). Furthermore, the within peatland position of the sampled cores are of importance in this context. This because the distance from the peatland margin, where the input of minerogenic water occur, will likely have bearing for the strength of the minerogenic water impact – strongest close to margins, and gradually weaker with increasing distances from the peatland margin.

The key message is indeed linked to the minerogenic water inputs, which is supported by previously published literature from the area, as discussed on page 7 of this document. We do not have information on fluxes of nutrients into the studied mires but would be interested in looking into those in future studies, to further support this, and other studies on mire-catchment interactions. Earlier studies, for example those by Vitt et.al. (1995) and Gorham (1991), have pointed out the importance of nutrient inputs for peat accumulation. Monitoring or accurately quantifying the magnitude and dynamics of such fluxes can then help to better understand C accumulation in mires.

Action: To support our interpretations further, and to provide a context to the conclusions, we have added the following:

Lines 179-181: *'Interestingly, N deposition can potentially promote peat decay in shrub-covered mires, with consequent peat C losses, due to decreased ericoid mycorrhizal activity in peatlands³⁵.'*

Lines 258-259: *'Our findings highlight the need of future research to evaluate the magnitude and dynamics of nutrient inputs to better understand C accumulation in mires.'*

With the following references added:

Vesala, R., Kiheri, H., Hobbie, E., van Dijk, N., Dise, N., Larmola, T. (2021). Atmospheric nitrogen enrichment changes nutrient stoichiometry and reduces fungal N supply to peatland ericoid mycorrhizal shrubs. *Science of the total environment*, 794:148737

Vitt, D.H., Bayley, S.E., Jin, T.L. (1995). Seasonal variation in water chemistry over a bog-rich fen gradient in continental western Canada. *Canadian Journal of Fisheries and Aquatic Sciences*, 52(3), 587-606.

5) Number of study mires and peat cores are unclear. There are 8 study mires (sites) of different age, but in Figure 3, there are only depicted results from 6 mires (as far as I can see). Kindly explain. In Materials and methods, you mention that 13 cores were oven-dried, and that 4 were freeze-dried: giving a total of 17 cores. However, as I understand your sampling approach (i.e., 2 peat cores from 7 mires, and 1 core from 1 mire), which gives in total 15 cores. Here I may have misunderstood something, but kindly explain.

We extracted peat cores from 8 different mires, but out of these we could only successfully apply the CF:CS or CRS model which was used to estimate accumulation rates in 6 mires, as described on lines 270-272. Validating the application of the dating models is part of the methods section, and in order to be transparent about the success of ²¹⁰Pb dating, we decided to keep it there. The number of oven and freeze-dried cores should indeed be 15 in total. We apologize for this and thank the reviewer for noting this.

Action: We updated the methods with the correct number of cores:

Lines 317-318: *'Discs from 11 cores were oven-dried at 60 °C for 96 h, while discs from four cores were freeze-dried⁵⁹.'*

Lines 338-341: *'The CF:CS model could be applied in four out of the total fifteen sampled peat cores. Of these, one core was a lawn and three hummocks. For five lawn cores and one hummock core, the CRS model was applied., meaning that at the end 10 out of 15 cores could be dated and included in the statistical analyses.'*

SPECIFIC AND MINOR COMMENTS + SOME SUGGESTIONS

Line 37: Add a reference after emissions.

Action: A reference was added after 'emissions': *Frolking, S. & Roulet, N. T. Holocene radiative forcing impact of northern peatland carbon accumulation and methane emissions. Glob. Change Biol. 13, 1079–1088 (2007).*

Line 39: Consider very (before slow) and add anoxic (after water-saturated).

Action: Revised accordingly.

Lines 39-42: *'The water table and its influence on oxygen availability is currently believed to be the main enabler of peat accumulation, where long-term (annual to millennial) C storage is a result of very slow mass loss in the deep and permanently water-saturated anoxic peat layer ('catotelm') characterized by slow anaerobic decomposition³.'*

Lines 40-43: Alternative phrasing suggestion: Importantly, C accumulation over an annual to decadal timescale first occurs by photosynthesis and plant growth in the surface of the biologically highly active surface zone (acrotelm), where decomposition rates are high.....

Action: We have rephrased this sentence.

Lines 42-45: *‘Importantly, C accumulation over an annual to decadal timescale first occurs in the unsaturated surface zone (‘acrotelm’), where decomposition rates are higher due to readily available oxygen as well as an often high proportion of poorly decomposed labile C in the peat.’*

Lines 45-46: Anoxic is wrong here! Change to more oxygenated conditions in the surface peat.

Action: Revised accordingly.

Lines 45-48: *‘The net annual atmospheric C exchange from mires is often very small and can occasionally shift from being sinks of atmospheric C to becoming sources, where reduced C retention is almost always related to periods with deeper groundwater levels and more oxygenated conditions in the surface peat^{4,5}.’*

Line 48: Insert that (before will).

Action: This has been added. The full sentence reads:

Lines 49-50: *‘In addition to the water table, nutrient availability is a critical factor in determining the type of peat-forming plant community that will be established⁶.’*

Lines 51-55: About Sphagnum mosses: Consider adding this reference: van Breemen, N. 1995. How Sphagnum bogs down other plants. – Trends in Ecology and Evolution, 10: 270–275.

Action: We agree that this is a good reference to include and we have added it to the following sentence.

Lines 53-55: *‘For example Sphagnum mosses are capable of promoting a shallow and stable water table through various hydro-physical feedbacks⁷, and this group of mosses are central for peat accumulation in high-latitude mires^{8,9}.’*

Line 58: Add by (before iron).

Action: This has been added. The full sentence reads:

Lines 58-61: *‘In minerogenic, i.e. groundwater-fed, mires (fens), which are the dominant mire type at high latitudes¹⁴, N fixation can be stimulated by the availability of weathering-derived nutrients from the catchment¹⁵, primarily through the supply of phosphorus, but also by iron and molybdenum¹⁶.’*

Line 61: The concept smaller is vague. I suggest something like: At a local mire spatial scale (or perhaps: At a single mire spatial scale).

Action: We agree that a more descriptive explanation would reduce the ambiguity. The rephrased sentence reads:

Lines 63-66: *‘At the local spatial scale of a single mire, the presence of microtopographic features, such as elevated hummocks and the more extensive, flatter lawns introduces additional complexity as these microtopographic features represent a spectrum of wetness that influences oxygen and nutrient availability within the mire¹⁷.’*

Lines 61-70: Here is it relevant to refer to the above-mentioned Ecology and Oikos papers about surface peat.

Action: We have added the suggested references to the following sentence.

Lines 71-73: *‘This variation in peat characteristics can result in significant differences in peat C and N accumulation between hummocks and lawns within the same mire¹⁸⁻²⁰.’*

Ohlson, M. & Halvorsen Økland, R. 1998. *Spatial variation in rates of carbon and nitrogen accumulation in a boreal bog. Ecology* **79**, 2745–2758.

Økland, R. H. & Ohlson, M. 1998. *Age-depth relationships in Scandinavian surface peat: A quantitative analysis. Oikos* **82**, 29–36.

Line 70: Consider substituting microforms with hummocks and lawns.

Action: We changed the word ‘microforms’ to ‘hummocks and lawns’ and also changed the word ‘cause’ to ‘affect’ in the sentence above to avoid the double use of the word ‘cause’.

Lines 68-73: *‘Seasonality in climate and magnitude of precipitation and evapotranspiration can also affect nutrient transport within the mire, causing differences in nutrient availability across microtopographic features, with hummocks accumulating nutrients relative to lawns under dry conditions, and the opposite occurring under wet conditions¹⁷. This variation in peat characteristics can result in significant differences in peat C and N accumulation between hummocks and lawns within the same mire^{18–20}.’*

Line 85: Consider substituting rebound with land-uplift.

Action: Both ‘rebound’ and ‘land-uplift’ can be used, but for the reader unfamiliar with the isostatic movements and how they affect the land surface, the word ‘land-uplift’ may be more illustrative. We have changed ‘rebound’ to ‘land-uplift’ in the two places it occurred:

Lines 97-99: *‘The unique landscape settings formed by post-glacial isostatic land-uplift allow us to disentangle the role of mire succession (i.e. landscape ageing), from geomorphological catchment controls on mire nutrient regimes and accumulation patterns.’*

Lines 276-278: *‘The region was covered by the 3 km thick Scandinavian Ice Sheet during the last glacial period⁵⁰ and, as a result, represents one of the areas globally with highest land-uplift, today at a rate of around 9 mm yr⁻¹⁵¹.’*

Lines 134-137: Elaborate by a few more words about how much (i.e., give numbers) CAR differed between hummocks and lawns.

Action: We have added the following sentence:

Lines 107-108: *‘Over the entire mire population, CAR was up to 75 g C m⁻² yr⁻¹ higher in lawns than in hummocks.’*

Lines 443 and 447: I guess M.O. should be M.Ö.

Action: Öquist is his name, and since we use Ö in the author list, this should naturally be repeated in the author contributions. The use of O vs Ö is a matter of journal and use of special letters. We have updated the author contributions with M.Ö.

Reference list: Carefully check and correct use of capital letters in publication titles.

Action: We have checked and corrected capital letters.

Kind regards, and best wishes from Mikael Ohlson

In addition to edits related to the editor's and reviewer's comments, we have made the following modifications:

1. Line 18: removed 'it'
2. Line 24-25: specified the mire age 'formed during the last millennia'
3. Line 25: added 'organic matter'
4. Line 33: added 'carbon' as keyword
5. Line 36: added 'sequester and'
6. Lines 37-38: added 'CO₂' and 'CH₄'
7. Line 38: replaced 'from' by 'of'
8. Lines 55-56: changed parenthesis for nitrogen (i.e. '(N)') to where nitrogen is first mentioned. Throughout the text, apart from the figures that should stand alone, we have replaced 'carbon' by 'C' and 'nitrogen' by 'N' to keep the use of acronyms constant
9. Line 63: removed the word 'greater'.
10. Line 71-74: rephrased the sentence: '*Seasonality in climate and magnitude of precipitation and evapotranspiration can also affect nutrient transport within the mire, causing differences in nutrient availability across microtopographic features, where hummocks typically get higher nutrient inputs during dry conditions, while lawns get a higher nutrient load during periods of high groundwater*¹⁷.'
11. Line 94: move the positions of the word 'peat' and changed from '100-150 years' to 'past century' to be consistent throughout the manuscript.
12. Line 135: added 'as nutrients'
13. Line 139: added '(GWL)' to the figure caption.
14. For figure 4 an explanation of the error bars is provided on Line 232-233: '... Error bars correspond to uncertainty of Clymo's model outcome.' The word 'rates' was also added to the caption.
15. Lines 241-242: rewording of the sentence to: '*Our results on lawns are consistent with the theory that C accumulation in northern mires is limited by a stronger influence of groundwater relative to rainwater*³⁶⁻⁴⁰.'
16. Line 243: added 'from'
17. Line 246: removed 'particularly from the upslope catchment area'
18. Line 253: removed the words 'of this work'
19. Line 262: added 'emphasizes'
20. Line 263: added 'highly'
21. Line 275: replaced 'fens' by 'mires'
22. Line 279: changed to capital letter of S in 'Baltic Sea'
23. Line 320: changed from 'seconds' to 'second'
24. Line 322: changed from 'in' to 'at'
25. Line 325: changed from 'past' to 'recent'
26. Lines 378-381: rewording of the sentence to: '*For each peat profile, the mass fractions were calculated based on the dry weight of peat. Mean mass accumulation rate (MAR) was multiplied by the mean C mass content to derive the mean C accumulation rate (CAR) over the past 100 years.*'
27. Lines 389-395: rewording of the sentence to: '*This is interesting because studies of CAR based on mires representing early successional stages, such as those considered in our study, are scarce due to the rare occurrence of young landscapes (< 5,000 years) with strong weathering gradients. Consequently, most studies of recent accumulation rates based on ²¹⁰Pb chronologies are based on ecosystems that have accumulated peat over much longer periods than the mires described here. The estimated NAR for the last hundred years was within the same range as the NAR in corresponding young landscapes in Finland⁴⁸ and in the northernmost parts of Sweden⁶⁵.*'

28. Lines 401-415: please see the entire paragraph for several rewordings.
29. Line 421: replaced '*applied for calculating*' by '*used to calculate*'. Included '*the*' in two spots earlier in the paragraph.
30. Line 438: rephrased 'However, it should be noted that mire...'
31. Subheading '*Catchment attributes*' starting at line 455: please check for several rewordings.
32. Line 527: we have added funding information from the Swedish Research Council FORMAS, which was used for the project, apart from the already mentioned FORMAS grants: '*2016-05275*'.
33. Included a data availability statement on Lines 523-526: '*The ²¹⁰Pb data, as well as environmental data used as explanatory variables that support the findings of this study have been deposited figshare*'

Spatial variation in C accumulation in a boreal peatland – from Ecology 79: 2745 – 2758.